# Seasonal and latitudinal variations in sea ice algae deposition in the Northern Bering and Chukchi Seas determined by algal biomarkers

**Chelsea Wegner Koch**[1]\*, **Lee W. Cooper**[1], **Catherine Lalande**[2], **Thomas A. Brown**[3], **Karen E. Frey**[4], **Jacqueline M. Grebmeier**[1]

**1** University of Maryland Center for Environmental Science, Solomons, MD, United States of America, **2** Amundsen Science, Université Laval, Québec, QC, Canada, **3** Scottish Association for Marine Science, Oban, Scotland, United Kingdom, **4** Graduate School of Geography, Clark University, Worcester, MA, United States of America

\* cwegner@umces.edu

**Data Availability Statement:** All relevant data are within the paper and its Supporting Information files.

## Abstract

An assessment of the production, distribution and fate of highly branched isoprenoid (HBI) biomarkers produced by sea ice and pelagic diatoms is necessary to interpret their detection and proportions in the northern Bering and Chukchi Seas. HBIs measured in surface sediments collected from 2012 to 2017 were used to determine the distribution and seasonality of the biomarkers relative to sea ice patterns. A northward gradient of increasing ice algae deposition was observed with localized occurrences of elevated $IP_{25}$ (sympagic HBI) concentrations from 68–70˚N and consistently strong sympagic signatures from 71–72.5˚N. A declining sympagic signature was observed from 2012 to 2017 in the northeast Chukchi Sea, coincident with declining sea ice concentrations. HBI fluxes were investigated on the northeast Chukchi shelf with a moored sediment trap deployed from August 2015 to July 2016. Fluxes of sea ice exclusive diatoms (*Nitzschia frigida* and *Melosira arctica*) and HBI-producing taxa (*Pleurosigma*, *Haslea* and *Rhizosolenia* spp.) were measured to confirm HBI sources and ice associations. $IP_{25}$ was detected year-round, increasing in March 2016 (10 ng m$^{-2}$ d$^{-1}$) and reaching a maximum in July 2016 (1331 ng m$^{-2}$ d$^{-1}$). Snowmelt triggered the release of sea ice algae into the water column in May 2016, while under-ice pelagic production contributed to the diatom export in June and July 2016. Sea ice diatom fluxes were strongly correlated with the $IP_{25}$ flux, however associations between pelagic diatoms and HBI fluxes were inconclusive. Bioturbation likely facilitates sustained burial of sympagic organic matter on the shelf despite the occurrence of pelagic diatom blooms. These results suggest that sympagic diatoms may sustain the food web through winter on the northeast Chukchi shelf. The reduced relative proportions of sympagic HBIs in the northern Bering Sea are likely driven by sea ice persistence in the region.

**Funding:** Financial support was provided by grants from the NSF Arctic Observing Network program (1204082,1702456 and 1917469 to J. Grebmeier and L. Cooper; 1204044,1702137 and 1917434 to K. Frey, https://www.nsf.gov/funding/pgm_summ.jsp?pims_id=503222) and NOAA Arctic Research Program (CINAR 22309.07_UMCES_Grebmeier, https://arctic.noaa.gov/) to J. Grebmeier and L. Cooper. Research cruises in 2012 and 2013 were part of the Hanna Shoal Ecosystem Study for the COMIDA project funded by the U.S. Department of the Interior, Bureau of Ocean Energy Management (BOEM), Alaska Outer Continental Shelf Region, Anchorage, Alaska (https://www.boem.gov/regions/alaska-ocs-region/alaska-ocs-region) under BOEM Cooperative Agreement No. M11AC00007 with The University of Texas at Austin as part of the Chukchi Sea Offshore Monitoring in Drilling Area (COMIDA) and the BOEM Alaska Environmental Studies Program (https://www.boem.gov/about-boem/alaska-environmental-studies), to PIs J. Grebmeier and L. Cooper. Additional funding support was provided to C. Wegner (Koch) by the North Pacific Research Board Graduate Research Award (https://www.nprb.org/), the Cove Point Natural Heritage Trust (http://www.covepoint-trust.org/) and the Chesapeake Biological Laboratory Graduate Education Committee (https://www.umces.edu/cbl). The funders had no role in study design, data collection and analysis, decision to publish, or preparation of the manuscript.

**Competing interests:** The authors have declared that no competing interests exist.

## Introduction

Sea ice supports a diverse community of microalgae (primarily diatoms), bacteria, metazoan grazers, heterotrophic and mixotrophic protists, viruses and fungi [1–4]. Sea ice associated (sympagic) algae grow on the underside and bottom few centimeters of sea ice and within brine channels during sea ice formation and eventually decline as sea ice melts [1, 5–8]. However, the precise contribution of sea ice algae to total primary production throughout the Arctic is poorly constrained owing to difficulties in measuring production in these communities [5] and to the overlap in habitat of sea-ice associated species [8]. Estimates of sea ice algae contributions to total primary production in the Arctic are widely variable, ranging from 4 to 26% in seasonally ice covered waters [9] and upwards of 50% in the central Arctic Ocean [5]. Observations of a phytoplankton bloom below melt ponds in the Chukchi Sea indicated that satellite-based estimates of chlorophyll biomass in areas of sea ice may be an order of magnitude too low [10]. The observation of nearly all algal export before complete ice melt in the Eurasian Arctic Ocean further reflects the underestimation by satellite sensor platforms [11]. It has been suggested that these ice algae blooms are an important early season source of food to pelagic grazers and benthic communities [6, 12–16]. Yet gaps remain in our understanding of the spatial and temporal variability of sea ice primary production in the Arctic and the impact on high latitude food webs. The application of biogeochemical methods to quantify and monitor sea ice algae contributions to pelagic and benthic food webs can be used to address these limitations associated with traditional field and satellite-based observations of sympagic production.

Highly branched isoprenoids (HBI) are a class of lipids with $C_{20}$, $C_{25}$ and $C_{30}$ hydrocarbon structures comprised of $C_5$ isoprene units unique to diatoms and can serve as species-specific biomarkers based on the number and position of double bonds [17, 18]. HBIs are produced by several commonly occurring diatoms genera including *Haslea*, *Pleurosigma*, *Navicula* and *Rhizosolenia*, but are limited to a small number of species within these taxa [17, 19, 20]. A small subset of these diatoms associated with Arctic sea ice produce a monounsaturated HBI, which has been termed the "Ice Proxy with 25 carbons", or $IP_{25}$ [18] (Fig 1). The detection of $IP_{25}$ is presumed to indicate the current or prior presence of sea ice and ice algal production at a given location. The physiological drivers that influence the synthesis of $IP_{25}$ or the specific sea ice and environmental conditions that stimulate its production are not fully understood and have yet to be synthesized in a laboratory setting [19, 21, 22]. HBI II (Fig 1), a $C_{25:2}$ alkane co-synthesized with $IP_{25}$ in Arctic sea ice, often occurs in larger relative abundances than $IP_{25}$ and has proven useful as an additional sea ice proxy [22–24]. HBI III (Fig 1), a $C_{25:3}$ alkane, is ubiquitous throughout the world's oceans and serves as an indicator of production in open water and marginal ice zones [17, 25, 26]. Several sea ice indices have been developed based on the relative proportions of $IP_{25}$ and other HBIs (or phytoplankton sterols) to estimate the relative proportions of sympagic versus pelagic production [27–29].

Nearly half of summer Arctic sea ice, based on the September minimum extent, has been lost since the start of satellite observations (1979-present) [30, 31]. Therefore, associated changes to ice algal production are to be expected. Trends in sea ice extent and duration are variable from year-to-year and throughout the Arctic [31]. Across the Pacific Arctic region (Bering, Chukchi and Beaufort Seas), sea ice break-up is occurring earlier and forming later, leading to younger and thinner sea ice annually with persistence declining by 9 to 30 days per decade over the satellite record [15, 31–33]. Two record low maximum winter extent periods for the Bering Sea occurred in 2018 and 2019, along with a record low summer minimum extent for the Chukchi Sea in 2019 [33–35]. Recent models suggest that annual sea ice duration in the Bering Strait could be reduced by an additional 20–36 days before 2050 and upwards of 60 days in the Eastern Siberian, Chukchi and Beaufort Seas [36]. On the continental shelf,

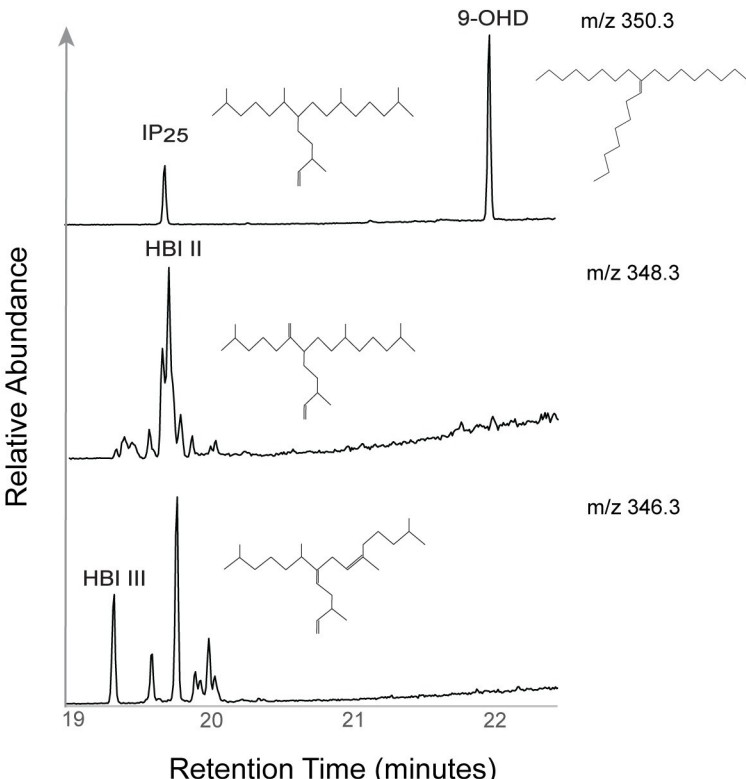

**Fig 1. Biomarker compounds and chromatograms.** The highly branched isoprenoid molecular structures for IP$_{25}$, HBI II, HBI III and the internal standard, 9-OHD. The compounds correspond with an example chromatogram from the surface sediment samples, showing the retention times and relative abundances.

August and September are essentially ice-free and the open water period is extending later into the fall.

Few HBI studies have been conducted on the productive shallow shelves of the Pacific Arctic marginal seas relative to the Eurasian and Canadian Arctic [22, 37]. Therefore, opportunities exist to improve our understanding of the dynamics of these biomarkers and their applications for ecosystem and paleoclimate studies. These measurements may also supplement existing knowledge from field-based and primarily satellite derived observations. The main goal of this study was first to establish the spatial distribution of IP$_{25}$, HBI II and HBI III from surface sediments throughout the region and investigate whether interannual variability can be distinguished. Additionally, there was a need to investigate the temporal dynamics of HBI production in the Pacific Arctic through biomarker fluxes (sediment traps). Finally, the fate or preservation of HBIs in this highly productive region was determined through measurements and comparisons of sediment cores collected from the biological hot spot on the shallow shelf relative to a deeper, less productive region on the Chukchi slope. By assessing the temporal and spatial dynamics of these biomarkers to establish a region-specific baseline, future studies may be able to employ this technique to monitor the rapid changes in sea ice occurring in the Bering and Chukchi seas.

## Regional setting

Currents in the Pacific Arctic region are dominated by a northward advection of water crossing the Bering shelf, converging in the Bering Strait and moving into the Chukchi Sea (Fig

2A). Different water mass components influence the transfer of associated heat content, organic matter and nutrients to the ecosystem [38–40]. There are three primary current pathways during the open water season: the nutrient-rich Anadyr Current to the west, Bering Sea water with summer and winter variants, and the warmer, nutrient-poor and seasonal Alaska Coastal Current to the east [38, 41, 42]. The northward flowing hydrography brings nutrient rich Pacific waters into the euphotic zone and supports persistent localized *in situ* production and advection and deposition of organic carbon to the benthos, and this productivity plays a role in the maintenance of benthic biological "hot spots" in the Bering Strait region [15].

The shallow shelf that spans from the northern Bering Sea to the northeast Chukchi Sea averages 40 meters in depth and has in recent years been seasonally ice covered for 0–3 months in the Bering Sea and 6–9 months in the Chukchi Sea [32]. The maximum median sea ice extent (1981–2010) has historically occurred in March in the northern Bering Sea and the minimum ice extent in September in the Chukchi Sea near the shelf break (Fig 2A). More recently, the minimum extent has shifted northwards away from the shelf break into the basin. The delayed freeze up in the Chukchi Sea ultimately impacts the winter sea ice extent and shifts the sea ice coverage in this entire region [36]. Throughout the sea ice cycle, primary production typically initiates with the ice algae bloom prior to sea ice melt, followed by or possibly partially seeding a pelagic phytoplankton bloom [8, 14, 43].

## Materials and methods

### Permitting

No national or international permitting was required as part of the sample collection efforts. Concerns regarding sampling in waters near Indigenous subsistence hunting areas was addressed by provision of cruise plans to the Arctic Waterways Safety Committee and some samples were imported into the United States from Canada using a US Fish and Wildlife Service Declaration for Importation or Exportation of Fish or Wildlife (USFWS Form 3–177).

### Sediment trap deployment

A sequential sediment trap (Hydro-Bios, Germany; 24 cups) was moored at 37 m depth, 8 m above the seafloor, as part of the Chukchi Ecosystem Observatory (CEO) located on the southeastern flank of Hanna Shoal (71.6˚N 161.5˚W, Fig 2B). The sediment trap was deployed in August 2015 and recovered in August 2016. Collection cups rotated at pre-programmed intervals ranging from one week during spring and summer to one month during winter. The last sample was excluded from the study as the sediment trap was recovered before the completion of the last rotation when the cup was still open. Before deployment, collection cups were filled with filtered seawater, adjusted to a salinity of 38 with NaCl to create a solution denser than ambient seawater to ensure material remained in the cup while open, and poisoned with formalin (4% final solution) to preserve samples during deployment and after recovery. In preservation tests of marine samples, formalin did not affect HBI proportions or indices relative to wet/dry freezing [44]. Trap samples were stored in the dark at room temperature until analysis, but we note that the effects of storage temperature on HBI degradation in formalin preserved samples have not been investigated [45].

### Diatom identification and quantification

Subsamples (0.1–3 mL) from the sediment trap bottles were adjusted to a volume of 3 mL with filtered seawater for the enumeration and identification of algal cells in an Utermöhl chamber [46]. A minimum of 300 phytoplankton cells were counted and identified to the lowest

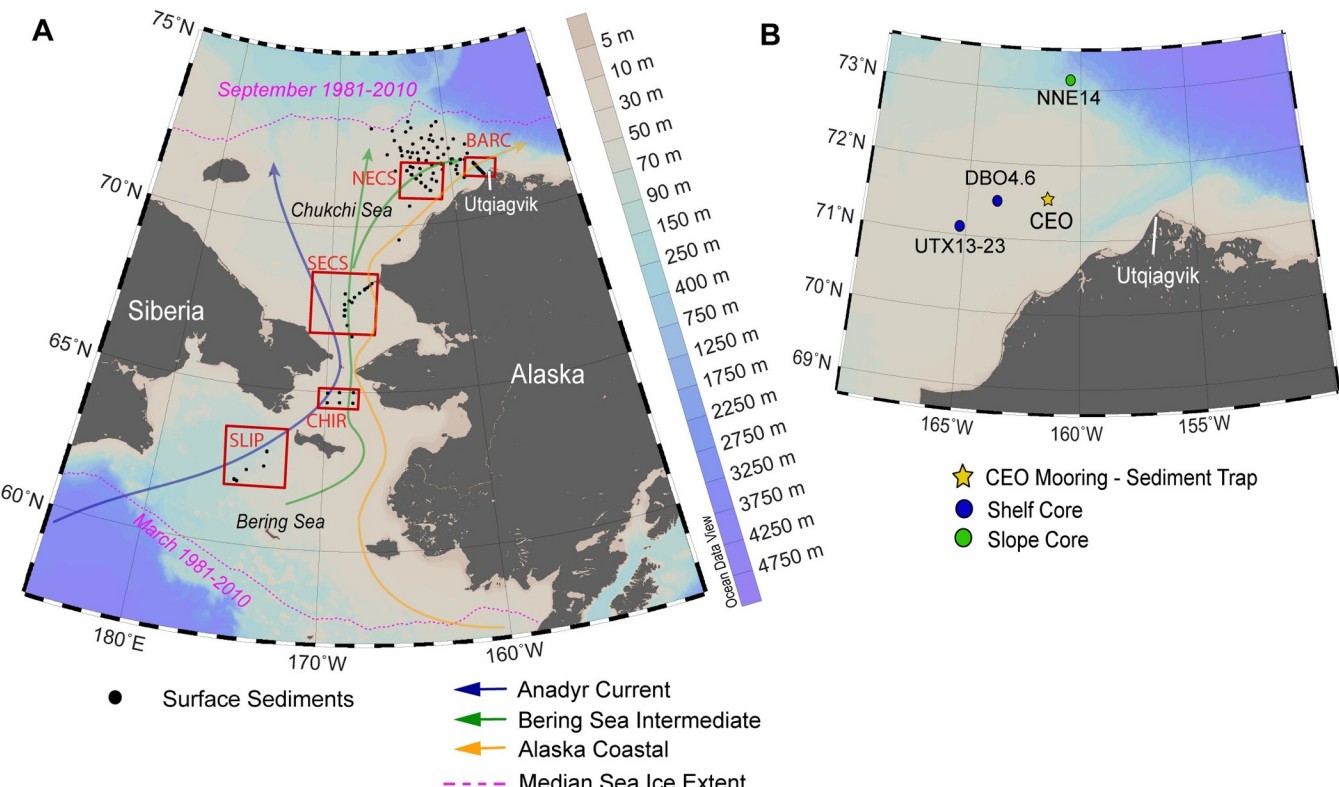

**Fig 2. Study site in the Pacific Arctic region. A)** The surface sediment sampling locations in the northern Bering and Chukchi Seas occurred within the framework of the Distributed Biological Observatory (DBO) regions (black boxes). The DBO regions in this study from south to north include: The St. Lawrence Island polynya (SLIP), Chirikov Basin (CHIR), southeast Chukchi Sea (SECS), northeast Chukchi Sea (NECS) and Barrow Canyon (BARC). **B)** The northeast Chukchi Sea region with the locations of the Chukchi Ecosystem Observatory (CEO) moored sediment trap and Haps core locations. Reprinted from Ocean Data View under a CC BY license, with permission from R. Schlitzer, original copyright 2020.

taxonomic level possible by inverted light microscopy at 100X, 200X or 400X depending on cell size using the Utermöhl method [46]. Empty algal cells (without chloroplasts) were distinguished from intact cells (with chloroplasts) assumed to be alive at the time of collection and resting spores [11]. Algal measurements were converted to daily fluxes depending on the subsampled volume and open cup duration of each sample.

Two sea ice exclusive diatom species, *Nitzschia frigida* (Grunow in Cleve and Grunow) and *Melosira arctica* (Dickie), were selected as indicators of the ice algae bloom. The *Gyrosigma/Pleurosigma/Haslea* group were selected to be the source of sympagic HBIs based on the currently known species that produce these lipids, which include *Pleurosigma stuxbergii var. rhomboides* (Cleve in Cleve and Grunow) Cleve, *Haslea kjellmani* (Cleve) Simonsen, *H. crucigeroides* (Hustedt) Simonsen, and *H. spicula* (Hickie) Lange-Bertalot [19, 47]. This broader group is not exclusively associated with sea ice. Another caveat is that *Pleurosigma* spp. includes species that produce the pelagic HBI III, including *P. intermedium* [48]. The diatom genera *Rhizosolenia* was selected as an indicator of the potential sources of HBI III. Species known to produce HBI III include *R. hebetata*, *R. polydactyla f. polydactyla* and *R. setigera* [20]. A more detailed analysis of the major diatom taxa and fluxes is discussed in Lalande et al. [49]

Subsamples for chlorophyll *a* (chl *a*) measurements were filtered onto GF/F filters (0.7 μm), extracted in 90% acetone for 24 h at -20˚C and measured on a Turner Design Model 10-AU

fluorometer following the methods outlined in Welschmeyer [50]. Samples were kept cool and in the dark prior to chl *a* measurements.

## Surface sediment collection

Surface sediment sampling was conducted on six annual expeditions from 2012 to 2017 on board the USCGC *Healy* (HLY; 2012, 2013, 2017) and the CCGS *Sir Wilfrid Laurier* (SWL; 2014, 2015, 2016). Sample collections from 2014 to 2017 were made at Distributed Biological Observatory (DBO) program sites (https://www.pmel.noaa.gov/dbo/), where long-term monitoring has been established in the Bering, Chukchi, and Beaufort seas [16, 51]. These sites are in the vicinity of five DBO long-term sampling station grids that were selected on the basis of having high productivity and/or biodiversity specifically in the north Bering Sea, the St. Lawrence Island polynya (SLIP), and the Chirikov Basin (CHIR), and north of Bering Strait, the southeast Chukchi Sea (SECS), the northeast Chukchi Sea (NECS) and Barrow Canyon (BARC) (Fig 2A). Sample collection in 2012 and 2013 focused primarily on the NECS region near Hanna Shoal (Fig 2B), but extended to all of the long-term Bering and Chukchi DBO benthic sampling sites in other sampling years. Surface sediments were collected by a van Veen grab ($0.1 \text{ m}^2$), with a trap door on the top that was opened prior to opening the grab in order to obtain relatively undisturbed sediments that were assayed for total organic carbon (TOC) and HBIs in the surface sediments. Samples were stored frozen (-20˚C) until analysis.

## Sediment core collection

Sediment cores were collected using a multi-HAPS corer (area = $133 \text{ cm}^2$) with stainless steel barrels and acrylic inserts deployed from the USCGC *Healy* in 2017 at station DBO 4.6 (71.62˚N 163.77˚W) and station NNE-14 (73.29˚N 160.04˚W, Fig 2B). A single core was collected at station DBO 4.6 on the shelf from a bottom depth of 43 m (Table 1). This core was sectioned shipboard for the first two centimeters at 1-cm intervals and the remaining length of the core at 2-cm intervals. A pair of cores were collected at station NNE-14 (1200 m depth; Table 1). Both cores were sectioned in 1 cm intervals at sea. Sections from core sections were immediately frozen and stored at -20˚C until analysis.

## Sediment core radiocesium measurements

The sectioned core from NNE-14 was analyzed for radiocesium ($^{137}$Cs) by gamma spectroscopy using a Canberra GR4020/S reverse electrode closed-end coaxial detector at the Chesapeake Biological Laboratory following established protocols [52]. Sedimentation data from another core collected in 2009 at station UTX13-23 (71.39˚N 166.28˚W), approximately 50 nautical miles from DBO4.6, was used in lieu of gamma analysis of the single core from DBO4.6 [52], which was instead used for analysis of IP$_{25}$ and other biomarkers. The $^{137}$Cs profile from core UTX13-23, which has been presented elsewhere [52] was used as a sedimentation proxy for DBO4.6, based on similarities in deposition [52]. DBO4.6 and UTX13-23 have similar grain sizes (50–75% $\geq$ 5 phi) and TOC (0.5–1%) [52], which have been found to be significantly correlated with radiocesium activity in surface sediments [53]. Additionally, we expected DBO4.6 to be highly influenced by bioturbation, as are most cores collected from this area of the Chukchi shelf [52]. This substitution was expected to be reasonable for the purpose of comparing cores collected in the biologically productive NECS region on the shelf relative to a core collected on the less productive continental slope (NNE-14).

**Table 1. Sediment coring locations and parameters.**

| Station | Deployment | Latitude °N | Longitude °W | Bottom depth (m) | Core length (cm) | Distance from CEO (nm) |
|---------|-----------|------------|-------------|-----------------|-----------------|----------------------|
| NNE-14 | 9/5/2017 | 73.33 | -160.17 | 1281 | 20 | 107 |
| DBO 4.6 | 8/31/2017 | 71.62 | -163.77 | 43 | 18 | 43 |
| UTX13-23 | 8/5/2009 | 71.39 | -166.28 | 46 | 16 | 92 |

Sediment core station names, collection dates, coordinates, station bottom depth, length of the Haps cores and distance from the Chukchi Ecosystem Observatory (CEO) mooring. All sediment cores were collected with a Multi-Haps stainless steel corer. Cores were collected from the northeast Chukchi shelf (DBO 4.6 and UTX13-23) and slope (NNE-14).

## Biomarker extraction

HBIs were extracted from surface sediment samples (n = 184; S1 Table), sediment trap sample cups (n = 23), and two sectioned sediment cores. Surface sediment and core samples were freeze dried for 48 hours, homogenized by mortar and pestle, followed by subsampling of approximately 1 g dried sediment. Sample cups from the sediment trap were gently mixed before subsamples were extracted with a modified pipette to enable the collection of larger particles for the measurement of HBIs. Sample volumes varied from 10 to 30 mL to accommodate the fluctuating particle flux through the year. These aliquots were filtered on Whatman GF/F filters (0.7 μm) and rinsed with deionized water. The filters were frozen overnight in petri dishes and placed into 8 mL vials for biomarker extraction.

HBIs were extracted following the methods of Belt et al. [54] and Brown et al. [29]. An internal standard (10 μL) of 9-octylheptadec-8-ene (9-OHD, 1 μg mL$^{-1}$) was added to the sample before extraction to facilitate yield quantification. Samples were first saponified in a methanolic KOH solution and heated at 70°C for one hour. Hexane (4 mL) was added to the saponified solution, vortexed, and centrifuged for 3 minutes at 2500 RPM for three iterations. The supernatant with the non-saponifiable lipids (NSLs) was transferred to clean glass vials and dried under a gentle $N_2$ stream to remove traces of residual methanolic KOH.

Elemental sulfur was removed from the sediment samples due to analytical interference with HBI III (*m/z* 346.3). This was accomplished by re-suspending the NSLs in 2 mL hexane with the addition of 1 mL of a tetrabutylammonium (TBA) sulfite reagent and 2 mL of 2-propanol. The solution was shaken for one minute and repeated, if necessary, until a precipitate formed. MilliQ water (3 mL) was added and the mixture centrifuged for 2 minutes at 2500 RPM. The hexane layer was removed into a clean vial with the hexane extraction and centrifugation repeated three times. The extract was dried under a gentle $N_2$ stream at 25°C and removed immediately once the solvent had evaporated.

Following sulfur removal, the extracts were re-suspended in hexane and fractionated using open column silica gel chromatography. The non-polar lipids containing the HBIs were eluted while the polar compounds were retained on the column. The eluted compounds were dried under $N_2$. 50 μL of hexane was added twice to the dried extract and transferred to amber chromatography vials.

## Biomarker analysis

The extracts were analyzed using an Agilent 7890A gas chromatograph (GC) coupled with a 5975 series mass selective detector (MSD) following methods outlined by Belt et al. [54]. Samples were analyzed on an Agilent HP-5ms column (30 m x 0.25 mm x 0.25 μm). The oven temperature was programmed to ramp up from 40°C to 300°C at 10°C/minute with a 10-minute isothermal period at 300°C. HBIs were identified using both total ion current (TIC) and

selective ion monitoring (SIM) techniques. TIC chromatograms and mass spectral output data were used to identify individual HBIs while SIM chromatograms were used to quantify the abundances by peak integration with ChemStation software. A purified standard of known $IP_{25}$ concentration was used to confirm the mass spectra, retention time and retention index (RI). Authentic HBI standards were also measured alongside the internal standard 9-OHD to determine the instrument response factor (RF, Table 2). For experimental purposes, samples were reanalyzed on an Agilent DB-5ms column (30 m x 0.25 mm x 0.25 μm) to determine the column-specific retention indices of these compounds. The HBIs were identified by their mass ions and RI including $IP_{25}$ (*m/z* 350.3), HBI II (*m/z* 348.3) and HBI III (*m/z* 346.3). To the best of our knowledge, the RIs for these HBIs have not been previously reported in the literature on a DB-5ms column (Table 2). A procedural blank was run every 9th sample.

Individual HBI concentrations in the surface sediment samples were normalized by TOC on an organic gram weight basis (S1 Table). TOC data from HLY12 (2012), HLY1702 (2017) and SWL 14–16 (2014–2016) cruises were accessed through the National Science Foundation's Arctic Data Center [55–59]. TOC data from the HLY13 (2013) cruise are available through another data archive, the National Oceanic and Atmospheric Administration National Centers for Environmental Information [60]. HBI concentrations from sediment trap samples were converted to daily fluxes depending on the subsampled volume and open cup duration of each sample and integrated over a 365-day period to annual fluxes.

The relative abundances of the sympagic HBIs ($IP_{25}$ and HBI II) to the pelagic HBI (HBI III), were quantified in order to determine the proportions attributable to different organic carbon sources. An HBI fingerprinting index, termed "H-Print", was used to estimate the relative organic carbon contributions of sea ice algae versus phytoplankton sources [29]. The H-Print method (Eq 1), is calculated using the relative abundances of $IP_{25}$, HBI II and HBI III, as determined by GC-MSD methods:

$$H-Print\% = \frac{HBI\ III}{\sum(IP_{25} + HBI\ II + HBI\ III)} x100 \tag{1}$$

The estimated organic carbon contribution resulting from the H-Print analysis varies from 0% to 100%, with lower values indicative of proportionally greater sympagic inputs and higher values indicative of proportionally lower sympagic inputs (i.e. substantial pelagic diatom sources). Analytical error from replicate control tests was determined to be less than 14% (relative standard deviation, RSD) for HBI quantification and less than 12% (RSD) for H-Print values.

## Sea ice concentration and snow cover

At the sediment trap location, daily averaged sea ice concentrations were retrieved at a 12.5-km resolution from the National Snow and Ice Data Center (NSIDC, https://nsidc.org) using the Defense Meteorological Satellite Program (DMSP) Special Sensor Microwave Imager/Sounder (SSMIS) passive microwave data. Snow depth on top of sea ice was retrieved at a 25-km resolution from the Northern Hemisphere snow depth files derived from the SSMIS data. Daily sea ice concentration and snow depth were averaged for a delimited region above the mooring (44 x 44 km; 71.4–71.8˚N; 161.4–161.9˚W).

The spring sea ice concentration (SpSIC) for each year of the study was averaged from monthly (April-June) sea ice concentration using DMSP SSMIS data [28]. The mean sea ice concentration at each of the sediment sample locations was extracted from the pixel containing the station location. The sea ice break-up dates were determined at each of the surface sediment sample locations. The sea ice break-up date was defined as the date when the pixel containing the station registered two consecutive days of sea ice concentration ≤15%, a common

**Table 2. HBI parameters for gas chromatograph-mass spectrometry.**

| Biomarker | m/z | Response Factor | Retention Index HP-5ms | Retention Index DB-5ms |
|---|---|---|---|---|
| IP$_{25}$ | 350.3 | 5 | 2081 | 2071 |
| HBI II | 348.3 | 12 | 2082 | 2075 |
| HBI III | 346.3 | 3 | 2044 | 2032 |

Individual biomarkers and the instrument response factors determined for this study. The known retention indices for the HP-5ms column were used for analysis and the RI for a DB-5ms column were experimentally reported.

threshold for open water conditions in sea ice studies [32]. The sea ice break-up date was then subtracted from the sample collection date to determine the ice-free period prior to sampling at each specific location of interest.

### Statistical analysis

Spatial analysis of the biomarker concentrations and H-Print values were conducted with ODV using DIVA (Data-Interpolating Variational Analysis) gridding methods [61]. All other statistical analyses were performed in R v. 3.6.1 [62] and plots were produced using the package ggplot2 [63]. Multiple linear regressions were used to investigate correlations between sea ice data and H-Print. One-way ANOVA testing and Tukey Honest Significant Difference (HSD) multiple pairwise comparisons were used to analyze the differences in relative HBI concentrations by DBO region. Principal components analysis (PCA) was used to analyze the impact of individual relative biomarker abundances at each location. Pearson product moment correlations were used to test for relationships among biomarker, diatom and chl *a* fluxes.

## Results

### Annual cycle of sea ice concentration, biomarker and diatom fluxes

At the CEO mooring site, open water conditions persisted from the initial deployment in mid-August through mid-November 2015 (Fig 3A). The increase in sea ice concentration in late November 2015 indicated a rapid sea ice freeze-up and the site remained ice-covered through mid-July 2016 (Fig 3A). Snowmelt first occurred in May 2016 and sea ice melt initiated in June 2016 (Fig 3A). Some sea ice (>15%) however remained present above the sediment trap until the end of deployment.

Chl *a* fluxes ranged from 1.5 to 1.9 mg m$^{-2}$ d$^{-1}$ from August through September 2015. Chl *a* levels remained relatively low (below 0.2 mg m$^{-2}$ d$^{-1}$) from December 2015 through April 2016. Chl *a* rapidly increased in late June 2016 and the maximum flux occurred in late July 2016 at 4.9 mg m$^{-2}$ d$^{-1}$ (Fig 3B). Similarly, POC fluxes were highest from August through September 2015 (1.09 to 1.18 g C m$^{-2}$ d$^{-1}$), a decline through the winter months and steady increase beginning in April 2016. The POC flux reached 1.04 g C m$^{-2}$ d$^{-1}$ in late July 2016 before the trap was recovered (Fig 3B).

The sympagic diatom fluxes are indicated by *N. frigida* and *M. arctica* (Fig 3C). *N. frigida* was first detected in the sediment trap in early April 2016, increased through late May 2016 and was no longer detected in early June 2016. *N. frigida* reappeared in mid-June and the maximum flux occurred in late June 2016. *M. arctica* was detected in the trap in early September 2015 and did not reappear until the maximum flux occurred in June 2016, corresponding to the peak flux for the exclusively sympagic species. *M. arctica* resting spores were present in August and September 2015, reappeared in May and remained consistently present until the end of the deployment. The *Gyrosigma/Pleurosigma/Haslea* group was detected in the trap

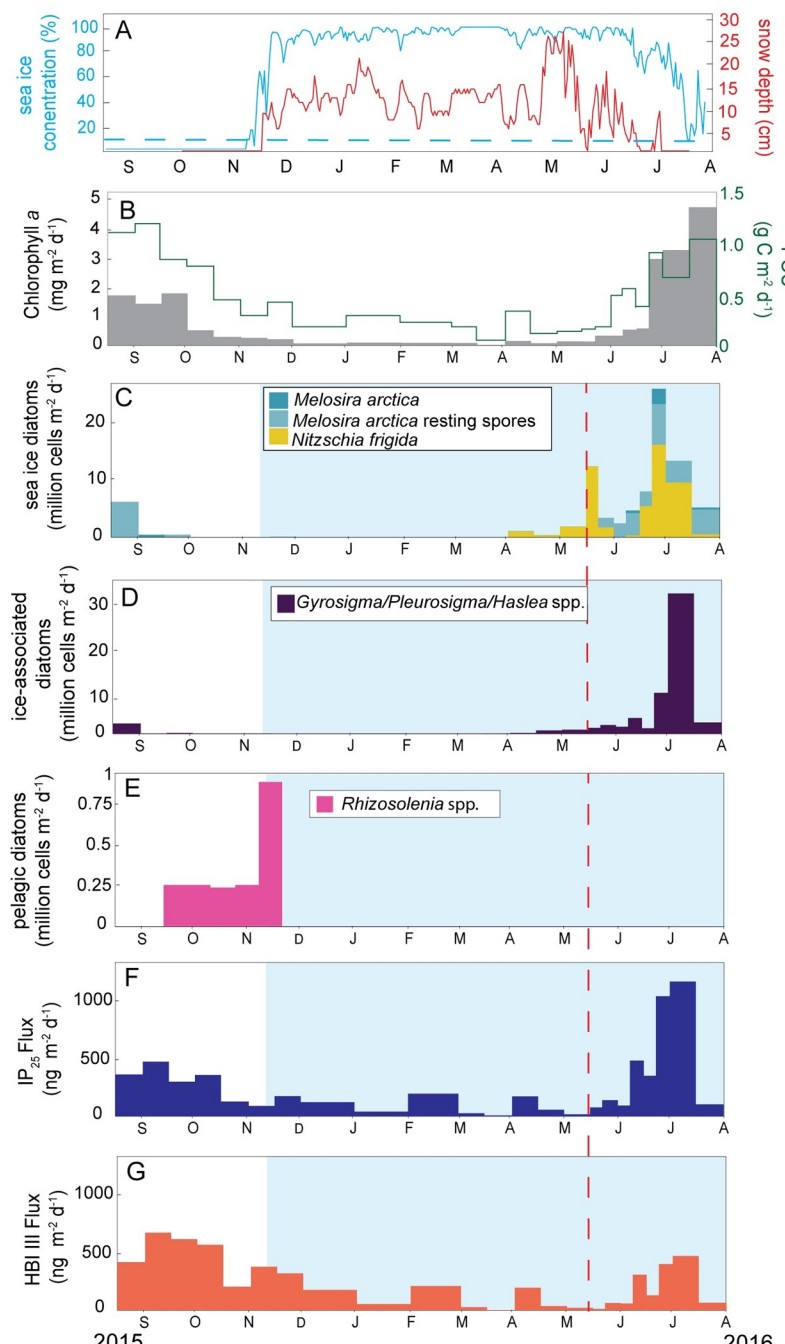

**Fig 3. Sea ice concentration, snow depth, and annual fluxes of diatoms and biomarkers at the Chukchi Ecosystem Observatory 2015–2016.** The parameters measured from the CEO sediment trap from August 2015 –August 2016 included: **A)** sea ice concentration (%) and snow depth (cm). The blue-dashed line indicates the 15% sea ice concentration threshold defining open water, **B)** chlorophyll *a* fluxes (mg m$^{-2}$d$^{-1}$) and POC fluxes (g C m$^{-2}$ d$^{-1}$). POC and chl *a* data from Lalande et al. 2020 [49] **C)** *Nitzschia frigida* and *Melosira arctica* fluxes (sea ice exclusive diatoms), **D)** *Gyrosigma/Haslea/Pleurosigma* fluxes (group containing HBI-producing species), **E)** *Rhizosolenia* spp. fluxes (group containing HBI III-producing species), **F)** IP$_{25}$ fluxes (ng m$^{-2}$d$^{-1}$), and **G)** HBI III fluxes (ng m$^{-2}$d$^{-1}$). All panels indicate the ice-covered period within the blue shaded boxes and the onset of snow melt is depicted by the red-dashed line.

throughout most of the year with the exception of early September through early November 2015 (Fig 3D). This group steadily increased starting in April 2016 and reaches a maximum in early July 2016. *Rhizosolenia* fluxes were only detected as intact cells from September through November 2015 (Fig 3E) although there were substantial fluxes of fragments year round (data not shown). The peak flux occurred in mid-November 2015.

$IP_{25}$ was detected throughout the entire sampling period (Fig 3F). $IP_{25}$ fluxes in the initial winter months (December 2015 through February 2016) occurred without the corresponding diatom groups recorded in the traps (Fig 3D and 3F). $IP_{25}$ fluxes began to increase in mid-May and reached a maximum in early July 2016 at 1331 ng m$^{-2}$ d$^{-1}$ (Fig 3F and Table 3). $IP_{25}$ sharply declined to 119 ng m$^{-2}$ d$^{-1}$ in late July (Table 3). This precipitous decline coincided with the peak chl *a* flux (Fig 3B). Overall, $IP_{25}$ fluxes mirrored the export of the *Gyrosigma/Pleurosigma/Haslea* taxonomic group. HBI III was also detected throughout the year (Fig 3G). The HBI III peak flux corresponded to the maximum *Rhizosolenia* spp. flux. HBI III fluxes reached a maximum flux of 799 ng m$^{-2}$ d$^{-1}$ in September 2015 (Fig 3G and Table 3). As indicated by the H-Print index, the sympagic diatom signal was present but low from September 2015 to late November 2015 with H-Print values ranging from 48–70% (Table 3), representing a mixed to pelagic diatom composition. H-Print values indicated a strong sympagic diatom signal in late March through late July 2016, with the strongest sympagic indicators during mid-May, late June and early July. The annual flux of $IP_{25}$ was 60 µg m$^{-2}$ yr$^{-1}$, HBI II fluxes were 278 µg m$^{-2}$ yr$^{-1}$, and HBI III fluxes reached 87 µg m$^{-2}$ yr$^{-1}$ (Table 3).

A Pearson correlation test was conducted on the assigned diatom groupings, chl *a* fluxes, and HBI fluxes (Table 4). The group containing sympagic-HBI producing species (*Gyrosigma/Pleurosigma/Haslea*) was strongly correlated with $IP_{25}$ fluxes ($r = 0.80$, $p<0.001$). The group containing pelagic-HBI producing species (*Rhizosolenia spp.*) was not significantly correlated with HBI III fluxes. Chl *a* was positively correlated with $IP_{25}$ fluxes ($r = 0.60$, $p<0.01$) and *Gyrosigma/Pleurosigma/Haslea* spp. ($r = 0.56$, $p< 0.01$). $IP_{25}$ and HBI III were also positively correlated ($r = 0.61$, $p<0.01$). $IP_{25}$ was positively correlated with the sea ice diatom flux (*N. frigida* and M. *arctica*, $r = 0.58$, $p<0.05$).

## Distribution and variation of biomarker deposition

$IP_{25}$ was detected in all of the surface sediment samples (Fig 4). Localized high concentrations occurred in the NECS and BARC regions in 2013 and 2017 and in the Chirikov Basin in 2016. $IP_{25}$ concentrations were generally higher ($>3$ µg g$^{-1}$ TOC) overall in the NECS and BARC regions relative to the lower latitude DBO regions. The SLIP region in 2015 was an exception with $IP_{25}$ concentrations reaching 12 µg g$^{-1}$ TOC at the SLIP3 station (S1 Table), which was the highest concentration observed of all years and stations. $IP_{25}$ data were only available for the SLIP region from 2015 through 2017, however, the concentration decreased over this time. Values exceeded 6 µg g$^{-1}$ TOC in four samples total (8–12 µg g$^{-1}$ TOC), which were determined statistically to be outliers by the IQR (Interquartile Range) method, and were incorporated as the maximum value (6 µg g$^{-1}$ TOC) rather than omitted for DIVA gridding. HBI III values were relatively consistent from year to year, with the highest concentrations found in the southeast Chukchi Sea (SECS) and northern Bering Sea (SLIP and CHIR) and minimal concentrations in the NECS (Fig 4).

The spatial distribution of H-Print index followed the general pattern of spring sea ice retreat each season, with weaker sympagic signatures (H-Print $> 60\%$) in the northern Bering Sea, particularly south of St. Lawrence Island and in the Chirikov Basin (Fig 5). The NECS and BARC regions displayed an elevated to moderate sea ice signal each year, with mean H-Print values ranging from ~21–59% for NECS and ~38–49% for BARC (Table 5). Spring sea ice

**Table 3. Sediment trap summary data.**

| Sampling Period | IP$_{25}$ Flux (ng m$^{-2}$d$^{-1}$) | HBI II Flux (ng m$^{-2}$d$^{-1}$) | HBI III Flux (ng m$^{-2}$d$^{-1}$) | H-Print (%) |
|---|---|---|---|---|
| 16–31 August 2015 | 413 | 2209 | 495 | 24 |
| 1–15 September 2015 | 540 | 2654 | 799 | 37 |
| 16–30 September 2015 | 341 | 1957 | 732 | 53 |
| 1–15 October 2015 | 408 | 1477 | 673 | 63 |
| 16–31 October 2015 | 146 | 496 | 242 | 63 |
| 1–15 November 2015 | 103 | 490 | 448 | 70 |
| 16–30 November 2015 | 199 | 873 | 380 | 49 |
| 1–31 December 2015 | 140 | 627 | 209 | 39 |
| 1–31 January 2016 | 47 | 220 | 64 | 37 |
| 1–29 February 2016 | 223 | 1143 | 250 | 29 |
| 1–15 March 2016 | 32 | 162 | 31 | 33 |
| 16–31 March 2016 | 10 | 44 | 2 | 18 |
| 1–15 April 2016 | 197 | 1067 | 232 | 29 |
| 16–30 April 2016 | 65 | 326 | 43 | 20 |
| 1–15 May 2016 | 20 | 100 | 25 | 26 |
| 16–22 May 2016 | 88 | 420 | 14 | 8 |
| 23–31 May 2016 | 160 | 641 | 75 | 11 |
| 1–7 June 2016 | 107 | 509 | 70 | 16 |
| 8–15 June 2016 | 550 | 2212 | 366 | 19 |
| 16–22 June 2016 | 400 | 1795 | 154 | 10 |
| 23–30 June 2016 | 1186 | 5500 | 476 | 9 |
| 1–15 July 2016 | 1331 | 6903 | 559 | 7 |
| 16–31 July 2016 | 119 | 650 | 78 | 12 |
| **Total Annual Flux** | **60 µg m$^{-2}$yr$^{-1}$** | **278 µg m$^{-2}$yr$^{-1}$** | **87 µg m$^{-2}$yr$^{-1}$** | |

Summary of daily (ng m$^{-2}$d$^{-1}$) and annual (µg m$^{-2}$yr$^{-1}$) HBI fluxes at the Chukchi Ecosystem Environmental Observatory moored sediment trap. IP$_{25}$ and HBI II are sea ice (sympagic) algae biomarkers and HBI III is a phytoplankton (pelagic) biomarker. The H-Print index represents the relative proportion of the pelagic to sympagic contribution of the total HBI flux. Low H-Print values indicate elevated sea ice algae contributions while high H-Print values indicate higher contributions of pelagic diatoms.

concentrations derived from satellite data indicated sea ice persistence through July in the northeast Chukchi Sea for 2012 to 2017 and an increasing open water period from the Chirikov Basin north to the southeast Chukchi sea from 2014 to 2017. By the spring months (April-June), the lower latitude stations were consistently ice-free. Four stations in 2017 had duplicate

**Table 4. Pearson product-moment correlation matrix for flux data.**

| | Sea ice diatom flux | *Gyrosigma -Pleurosigma- Haslea* flux | *Rhizosolenia* flux | Chlorophyll *a* flux | IP$_{25}$ flux |
|---|---|---|---|---|---|
| *Gyrosigma/Pleurosigma/Haslea* flux | **0.58** | | | | |
| *Rhizosolenia* flux | -0.25 | -0.16 | | | |
| Chlorophyll *a* flux | **0.55** | **0.56** | 0.13 | | |
| IP$_{25}$ flux | **0.73** | **0.80**[*] | -0.09 | **0.60** | |
| HBI III flux | 0.11 | 0.26 | 0.31 | 0.35 | **0.61** |

The Pearson product-moment correlation coefficients for the sediment trap flux parameters including: sympagic diatom flux (*N. frigida* and *M. arctica*), *Gyrosigma/Pleurosigma/Haslea* spp. flux, *Rhizosolenia* spp. flux, chlorophyll *a* flux, IP$_{25}$ and HBI III fluxes. Values in bold indicate significant correlation ($r$) where $p < 0.05$. An asterisk indicates targeted associations for HBI and diatom comparisons. Sample sizes for all parameters were n = 23.

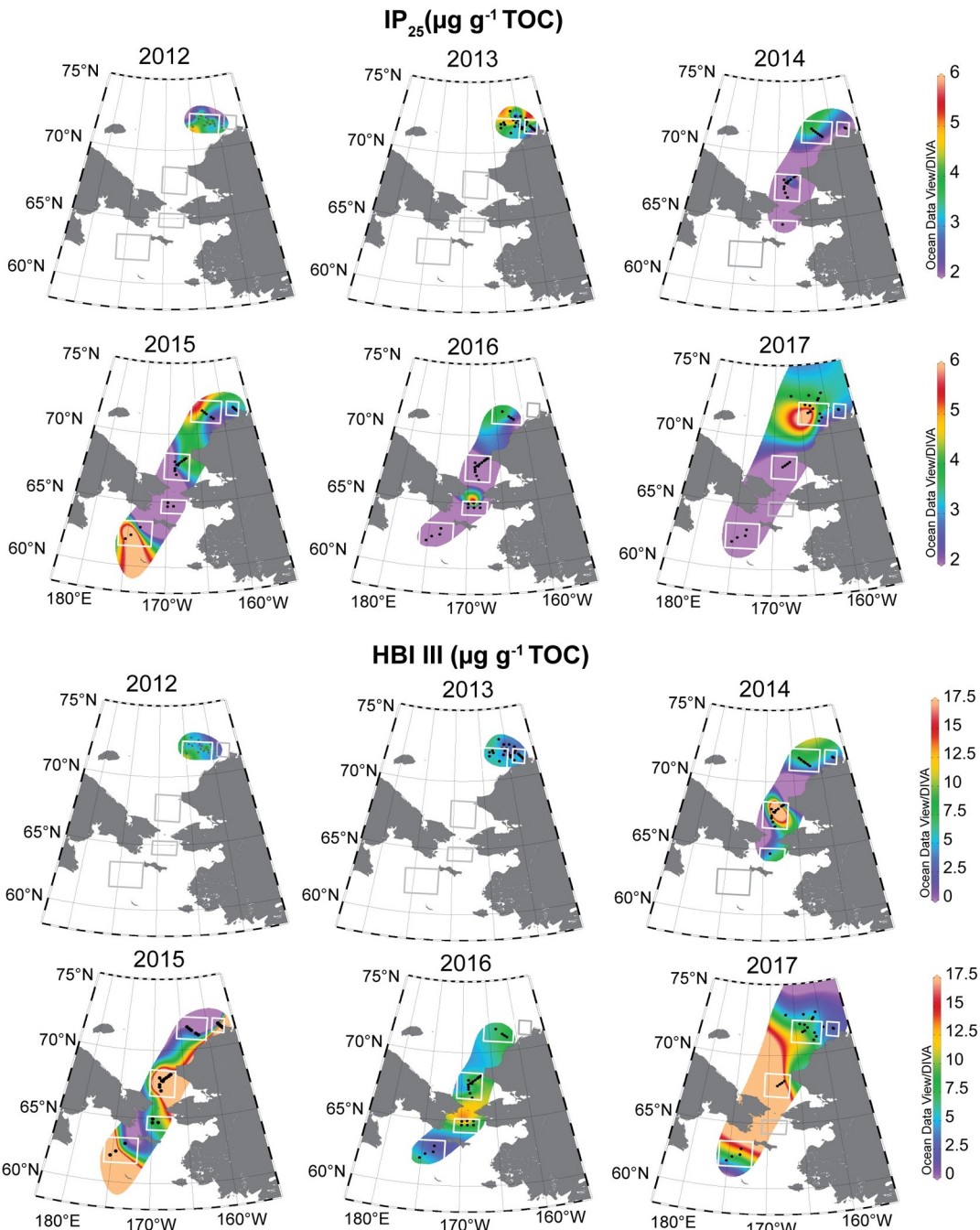

**Fig 4. IP$_{25}$ and HBI III biomarker distributions.** Spatial distribution of the relative abundances of IP$_{25}$ and HBI III concentrations (µg g$^{-1}$ TOC) in surface sediments from 2012–2017. The white and grey bounding boxes indicate the DBO regions from south to north (SLIP, CHIR, SECS, NECS and BARC). Not all sampling stations and DBO regions were able to be occupied every year due to sea ice or weather, indicated by grey boxes (no data collected). IP$_{25}$ and HBI III values were used as sympagic and pelagic diatom proxies, respectively, for the H-Print analysis. Reprinted from Ocean Data View under a CC BY license, with permission from R. Schlitzer, original copyright 2020.

surface sediment samples from Haps core tops and Van Veen grabs. The maximum difference in H-print was 6%, within the margin of error (12%).

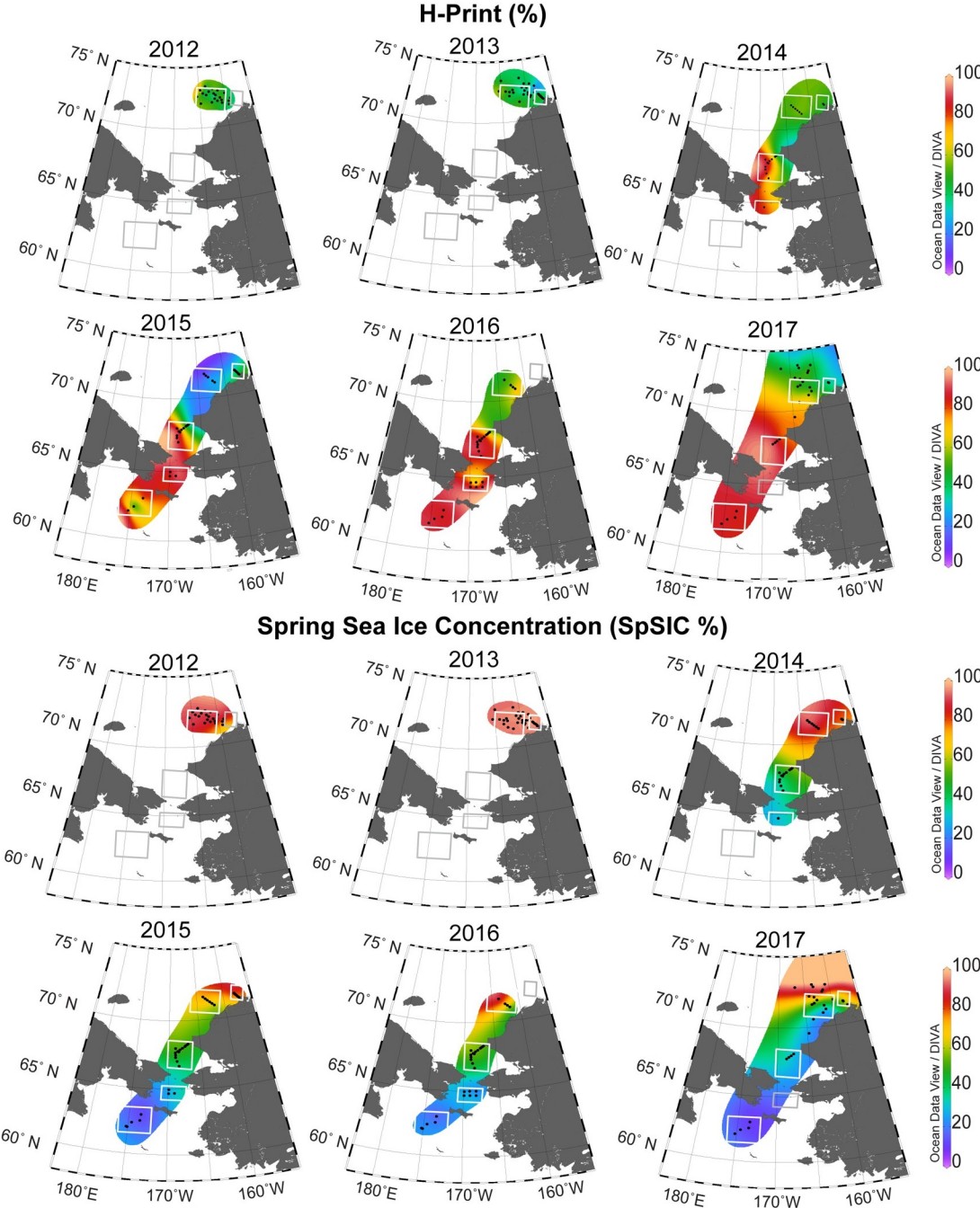

**Fig 5. H-Print index and satellite-derived sea ice concentration.** The spatial distribution of H-Print (%) in surface sediments from 2012–2017 and the spring sea ice concentration (SpSIC%) derived from April–June mean sea ice concentrations collected from SSMIS passive microwave data (NSIDC). The white and grey bounding boxes indicate the DBO regions from south to north (SLIP, CHIR, SECS, NECS and BARC). Not all sampling stations and DBO regions were able to be occupied every year due to sea ice or weather, indicated by grey boxes (no data collected). H-print ranges from 0–100%, where low values indicate elevated sea ice algae contributions while high values indicate higher contributions of pelagic diatoms. Reprinted from Ocean Data View under a CC BY license, with permission from R. Schlitzer, original copyright 2020.

To assess the relationship between the H-Print index and sea ice, linear regressions of two sea ice metrics were examined, including the SpSIC and sea ice break-up date relative to sample collection (Fig 5). Both relationships were significant at the 99% confidence level but the

**Table 5. Regional summary of H-Print sea ice index spatial distributions.**

| DBO Region | Mean H-Print (%) by Year | | | | | |
|---|---|---|---|---|---|---|
| | 2012 | 2013 | 2014 | 2015 | 2016 | 2017 |
| St. Lawrence Island Polynya (SLIP) | - | - | - | 71 ± 32 (5) | 88 ± 2 (5) | 86 ± 1 (4) |
| Chirikov Basin (CHIR) | - | - | 82 (1) | 88 ± 6 (4) | 82 ± 18 (6) | - |
| Southeast Chukchi (SECS) | - | - | 76 ± 16 (12) | 71 ± 24 (14) | 83 ± 11(14) | 87 ± 9 (7) |
| Northeast Chukchi (NECS) | 49 ± 9 (21) | 46± 7 (30) | 54 ± 3 (6) | 21 ± 10 (6) | 59 ± 8 (4) | 49 ± 11 (18) |
| Barrow Canyon (BARC) | - | 41 ± 9 (10) | 49 ± 6 (3) | 40 ± 22 (10) | - | 37± 4 (4) |

Mean H-Print (mean ± SD) by DBO region and year of sample collection. Sample sizes (n) are in parentheses.

SpSIC relationship was a better fit ($R^2$ = 0.46 versus $R^2$ = 0.34, n = 184). The SpSIC and H-Print regression shows that the locations with ice-coverage through spring had more substantial sympagic HBI contributions (Fig 6A). The number of ice-free days before sampling shows longer relative periods of open water were associated with reduced sympagic and elevated pelagic organic matter inputs (Fig 6B). There was a linear gradient and association between higher H-Prints and extended open water periods (lower spring sea ice concentration) at lower latitudes and lower H-Prints with higher spring sea ice and shorter ice-free periods at higher latitudes. There was a large degree of variability in both relationships.

To further explore the relationship between latitude and H-Print, the H-Print values were grouped by DBO region and plotted by latitude (Fig 7A). The box-and-whisker plots show the transition of increasing sea ice algal signature from south to north. There is also a greater degree of variability in the Chukchi Sea regions (SECS, NECS and BARC). The principal components analysis with individual HBIs ($IP_{25}$, HBI II, and HBI III) and grouped by DBO region also depict a divergence between the SLIP-CHIR-SECS and the NECS-BARC regions.

A one-way ANOVA test for the H-Print values grouped by DBO region suggests that the mean values were statistically different ($p<0.001$, $F$-value = 55.97). A Tukey multiple-pairwise comparison indicates that the differences between NECS-BARC, SECS-CHIR, CHIR-SLIP, and SLIP-SECS were not significant. In other words, the northern regions (NECS, BARC) are

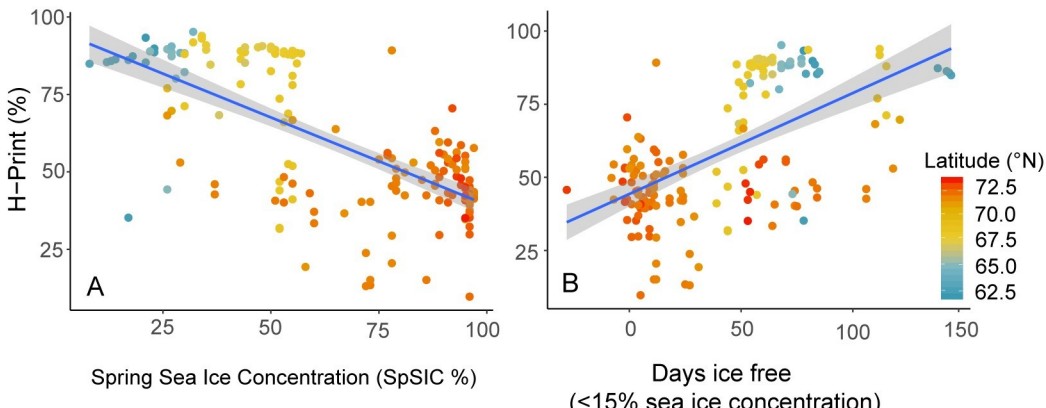

**Fig 6. Latitudinal variation and correlation of the H-Print index with sea ice.** The 2012–2017 H-Print values were compared with two different metrics for sea ice to determine the influence on the biomarkers. **A)** Linear regression of H-Print and the mean Spring Sea Ice Concentration (SpSIC) derived from April-June monthly sea ice concentration values. **B)** Linear regression of H-Print and the ice free period determined by the sea ice break-up date relative to sample collection date. Both relationships are shown with respect to latitude.

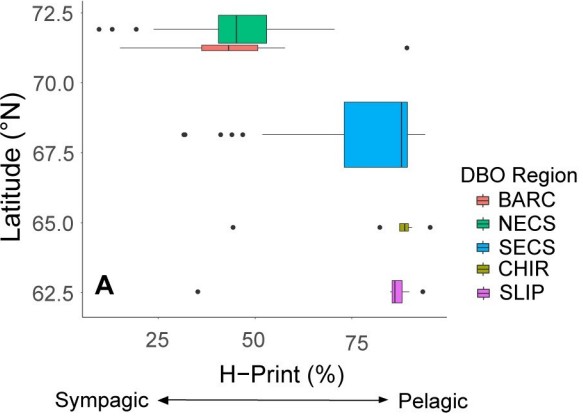

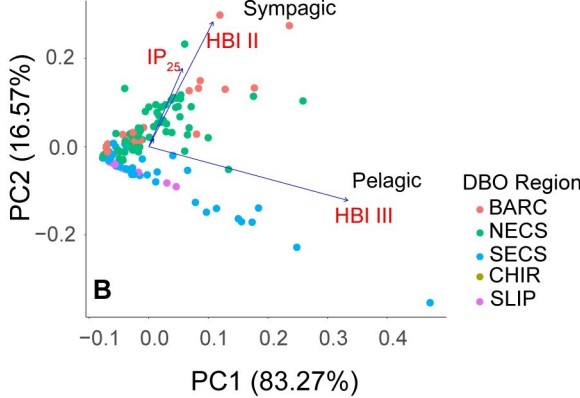

**Fig 7. H-Print index by DBO region.** Statistical analysis of the H-Print values from surface sediments in relation to the location **A)** boxplot of H-Print variability by DBO region and latitude **B)** Multivariate separation of surface sediments visualized by principal components analysis (PCA) of individual HBIs (IP$_{25}$, HBI II isomers, and HBI III) grouped by DBO region.

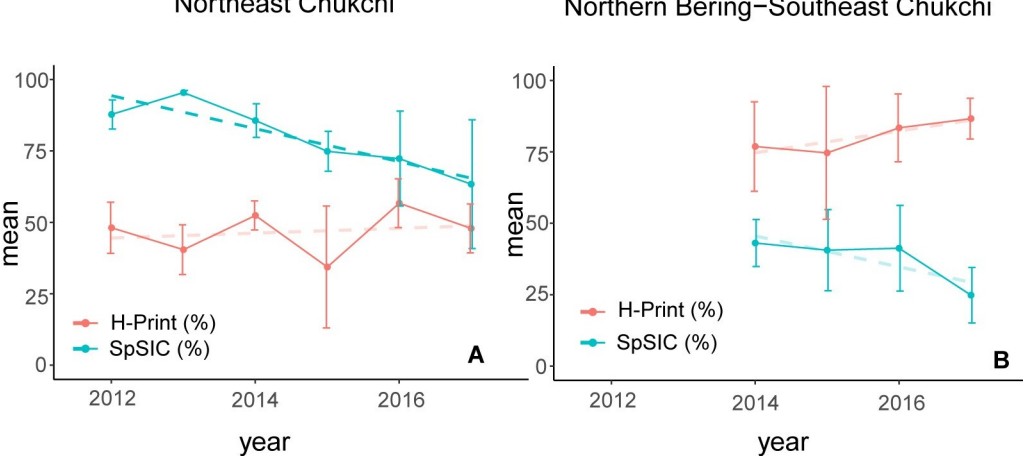

**Fig 8.** Annual Trends in H-Print and Spring Sea Ice Concentration **(A)** the northern Chukchi DBO regions (NECS and BARC) for 2012–2017 and **(B)** the Bering-southeast Chukchi DBO regions (SLIP, CHIR and SECS) for 2014–2017. The bold dashed line shows the only significant trend ($p < 0.01$).

similar to each other and the southern regions (SLIP, CHIR, SECS) are similar to each other, but both of the northern stations differ from each of the southern stations ($p<0.001$). The H-Print index varied by latitude, with the greatest amount of variability among NECS and BARC locations in addition to a stronger sea ice carbon signature at the higher latitudes (71–73˚N) and stronger pelagic influence at the lower latitudes (62–68˚N; Fig 7A). The PCA of the relative abundances of individual HBIs grouped by DBO region also supports this divergence in H-Print between the northern Bering and northeast Chukchi Seas (Fig 7B). The first principal component (PC1) accounted for 83.3% of the variation, with primary contributions from HBI III, and the second principal component (PC2) accounting for 16.6% of the variation, with HBI II and HBI III as the primary contributors (Fig 7B).

The annual mean H-Print and SpSIC values for the two distinct regions were grouped to assess temporal trends over the study period (Fig 8). Based on regression analyses, the only significant trend identified was for the SpSIC in the northeast Chukchi Sea, with a decline of 5.8% per year ($p<0.001$). However, the patterns are consistent for both regions, where the SpSIC is declining and the H-Print is increasing from 2012–2017 in the northeast Chukchi (Fig 8A) and from 2014–2017 in the northern Bering and southeast Chukchi Seas (Fig 8B).

### HBI profiles in sediment cores

The core collected on the Chukchi shelf break, NNE14, showed minimal signs of bioturbation, based visually on three distinct layers of sediment and validated by $^{137}$Cs measurements indicating a single subsurface peak in the upper 5 cm (Fig 9) that can be interpreted as corresponding to the bomb fallout peak in 1963 [52]. The top 3 cm of the core consisted of oxidized red-brown sediment, the next 5 cm consisted of brown sediments with similar consistency as the shelf sediments, and the remaining length of the core was composed of grey, fine-grained sediments. The H-Print values for this core were generally homogenous and less than 30%, indicating a high and consistent degree of sympagic organic carbon contributions (Fig 9). The core collected on the Chukchi shelf, DBO 4.6, on the other hand, was subject to significant bioturbation, including by polychaete worms present in the core when it was sectioned (sometimes spanning multiple core intervals). The H-Print values from this core were higher than the slope core, with values ranging from 30–55%, representing a greater pelagic contribution compared to the slope core but still having substantial sympagic inputs.

## Discussion

Recent trends in sea ice formation and retreat in the Pacific Arctic include delayed freeze up in the Chukchi Sea, driven by increasing sea surface temperatures, water column heat content and atmospheric dynamics, which ultimately result in later ice formation and earlier retreat in the Bering Sea [32, 34, 64]. These recent higher surface water temperatures, particularly if paired with southerly winds in the winter, lead to conditions where sea ice does not reach the historical (1981–2010) median extent. In particular, the 2017–18 overwinter period was an extreme year for sea ice decline in the northern Bering Sea [33, 34, 64]. In 2018, the winter sea ice extent in the Bering Sea was the lowest on record followed by 2019, which was the second lowest maximum extent on record [33, 35]. These areas once recurrently covered by ice in winter and early spring were open waters. In July 2019, the Chukchi Sea also experienced record low sea ice extent, with sea ice retreating off of the shelf by this time [35]. Four of the six years analyzed in this study in the Chukchi Sea were among the top ten record low sea ice years based on regional analysis of satellite data [35]. Therefore, all of the data examined in this study have occurred in a period of anomalies in the overall record, or a new norm relative to the satellite record.

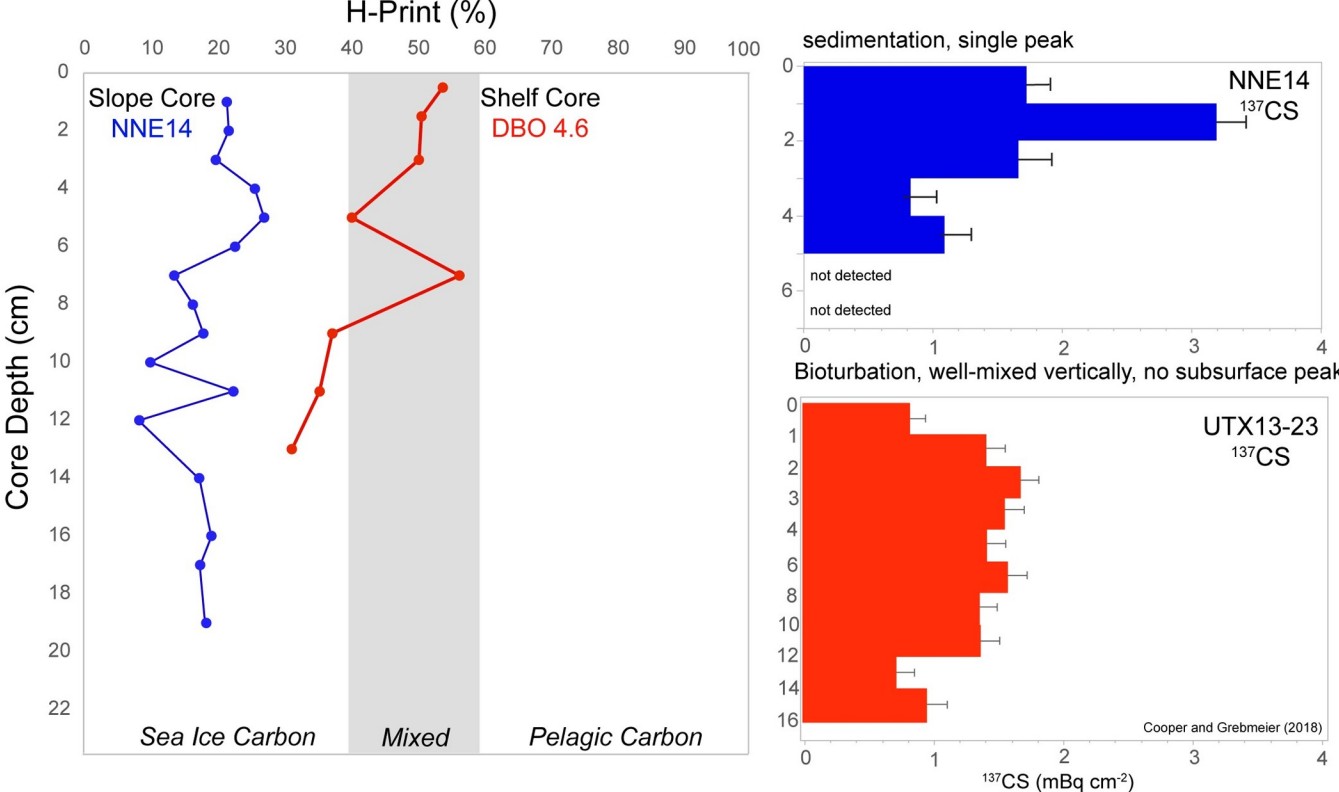

**Fig 9. H-Print and radiocesium profiles in sediment cores.** H-Print profiles for a core collected on the Chukchi slope, NNE-14 (blue), and a bioturbated core collected on the shallower Chukchi shelf at DBO 4.6 (red). $^{137}$Cs profiles for NNE-14 and UTX13-23, a core collected in close proximity to DBO 4.6, depict the consistent sedimentation or bioturbation of deposited material.

### Seasonal variations of HBI and diatom export in the northeast Chukchi Sea

The IP$_{25}$ and sea ice diatom fluxes observed at the CEO indicated an early summer sea ice algal bloom on the Chukchi shelf. Peak IP$_{25}$ fluxes during early July 2016 (1331 ng m$^{-2}$ d$^{-1}$), coincided with the largest flux of the diatom group *Gyrosigma/Pleurosigma/Haslea* (Fig 2D), which account for ~1% of the relative abundance of major diatom taxa groups [49]. This value is higher than maximum values observed in August 2008 and August 2009 in the Chukchi Borderland (46 ng m$^{-2}$ d$^{-1}$ and 33 ng m$^{-2}$ d$^{-1}$, respectively) [37]. This is not surprising given that generally shallow Arctic shelves are more productive than the basin and slope regions [11, 14]. The presence of sea ice associated species, such as *N. frigida* and *M. arctica*, provided additional indicators of ice algae release. Export of *N. frigida* was first detected at the CEO sediment trap in early April 2016 (Fig 3C), which was followed by the highest detected relative abundance of *Gyrosigma/Pleurosigma/Haslea* (~5%) in late April [49]. The slight increase of IP$_{25}$ fluxes from January to February 2016 (47 to 223 ng m$^{-2}$ d$^{-1}$, Table 3) corresponds to the reappearance of the *Gyrosigma/Pleurosigma/Haslea* taxonomic group [49]. This timing would correspond to the first seasonal deposition by known IP$_{25}$ producers contributing to the algal flux from the CEO site. This period also corresponds to an increase in HBI III (220 to 1143 ng m$^{-2}$ d$^{-1}$). The sharp decline in IP$_{25}$ in late July, along with an increase in pelagic diatom species (*Chaetoceros* and *Thalassiosira spp.*) [49], likely signified the end of ice associated diatom export as a result of the bottom few centimeters of sea ice melting that contain the most organic material and nearly all sea ice algae [7].

There were limitations in this study for comparing the selected diatom fluxes with the HBI fluxes, as taxonomy to the desired level was not feasible. The flux of the *Gyrosigma/Pleurosigma/Haslea* group into the sediment trap was used to compare the IP$_{25}$ fluxes, given the potential inclusion of the three or four known species that produce IP$_{25}$ from the genera *Haslea* and *Pleurosigma* [19]. However, this genus cluster also includes species that are not HBI producers and diatoms that are not considered exclusively sympagic. HBI-producing species are a minor taxa (ca. 1–5%) and only represent a small fraction of the abundances observed in this group [19]. Therefore, these results are presented with caution in regards to being the direct source for IP$_{25}$. However, the onset of increasing levels of IP$_{25}$ strongly corresponded to the increasing levels of the sympagic diatoms and the *Gyrosigma/Pleurosigma/Haslea* group as indicated by the Pearson product-moment correlation ($r = 0.73$ and 0.80, $p<0.001$; Table 4). The strong correlation of IP$_{25}$ with the sea ice diatom group and the *Gyrosigma/Pleurosigma/Haslea* genus group strengthens our interpretation that IP$_{25}$ is an appropriate sea ice proxy on the Chukchi shelf. In an example that echoes the complexities we observed in HBI source attribution, in a study conducted in an ice-covered fjord in Greenland, all known HBI-producing species were detected in ice cores and algal fluxes at 37 m but IP$_{25}$ production could only be attributed to *H. spicula* [47]. Limoges et al. [47] also found that *H. crucigeroides* and *H. vitrea* were producing both the diene (HBI II) and triene forms (HBI III), meaning that it is unclear what promotes synthesis of IP$_{25}$, including the sea ice conditions and other parameters that may promote or depress the synthesis of this compound. A recent study found that sea ice diatoms increase HBI concentration up to ten-fold when nutrients are a limiting factor [65]. We had to address similar uncertainties in determining the HBI III producing species in this study. The decision to investigate the correlations between *Rhizosolenia* fluxes with HBI III was made as this genus contains several HBI III producing species, but again would not encompass all potential sources. However, we found no correlation between the *Rhizosolenia* spp. and HBI III flux (Table 4).It is also noteworthy that the HBI III flux decreased in late July as chl *a* reach a maximum flux during a pelagic under-ice phytoplankton bloom (Fig 3B and 3G). There was also a positive correlation between IP$_{25}$ and HBI III fluxes that we cannot unambiguously interpret. This correlation could be due to the potential overlap of the taxa and broad assignments of possible biomarker producers (i.e. *Pleurosigma*). The weaker correlations with HBI III overall may suggest that HBI III perhaps is not reliable as a pelagic productivity indicator at this location and that *Rhizosolenia* did not adequately capture the source of the HBI III. This finding was also discussed in a recent study in the sea ice, raising similar complexities in assigning this to a pelagic source or the potential of regional implications [66]. An association of HBI II and HBI III was also observed in the Antarctic where it was suggested that HBI III is an indicator of the intense phytoplankton blooms that emerge in the marginal ice zone (MIZ) rather than open water, meaning that HBI III was more suitable as a proxy for sea ice seasonality or MIZ duration [67].

The HBI fluxes obtained at the CEO suggest that a combination of processes including production, resuspension and advection led to the persistent IP$_{25}$ signature in the algal flux recorded on the northeast Chukchi Sea shelf. The high export event at the end of June and beginning of July in 2016 (Fig 3B–3G) corresponded to the declining sea ice concentration rather than the early snowmelt in May (Fig 3A). This could also mean that IP$_{25}$ producing diatoms were present below the ice, rather than within the ice matrix. While pelagic diatoms were largely responsible for the chl *a* signal in the NECS region from August to October 2015, based upon taxonomic analysis, IP$_{25}$ and HBI II fluxes observed under-ice during April and May 2016 reflected a large proportion of sea ice algae in diatom export. Photosynthetically active radiation (PAR) measured at 33 m depth at the CEO began to increase ($>1$ uE cm$^{-2}$ s$^{-1}$) in March 2016, reaching upwards of 15 uE cm$^{-2}$ s$^{-1}$ in May 2016 [68]. The onset of increasing

PAR, along with snow melt (Fig 3A), triggered the initiation of the sea ice algae export, as reflected in the HBI fluxes and the first maxima of the sympagic diatom (*N. frigida*) in the trap material in May 2016. Export occurring prior to melt events was possibly due to the detachment of ice algae by currents and/or grazing processes. Diatoms that may have been incorporated into the ice matrix seeding a phytoplankton bloom (e.g. *Fragilariopsis* spp., *Pseudonitzschia/Nitzschia* spp.) dominated algal fluxes in July before complete sea ice retreat, along with the exclusively pelagic diatom *Chaetoceros* spp. [49]. A Bering Sea study previously found a fluid reciprocity between sympagic and pelagic diatoms through the melt season, with both groups incorporated into the sea ice matrix and a gradual transition of assemblages throughout the season [8].

The most notable finding from the sediment trap analysis was the detection of $IP_{25}$ fluxes year round, likely the result of both new production and resuspension events. Our observations are consistent with continuous fluxes of organic matter, which were recorded under land-fast ice from winter through late spring on the Mackenzie Shelf of the Beaufort Sea, although particulate organic carbon fluxes in winter were not consistent with diatom export [7]. In our samples, there was a lack of chloroplast-containing *Haslea* spp. during most of the winter months (October through February; Fig 3D). The winter sympagic HBI signal may be the result of resuspension, as supported by the low export of diatoms with chloroplasts recorded during winter [49].

Pelagic HBI III fluxes increased from 495 to 799 ng $m^{-2}$ $d^{-1}$ from August to September 2015, reflecting the export of an autumn phytoplankton bloom and/or resuspension due to storm activity (Table 2). The large flux of HBI II (2654 ng $m^{-2}$ $d^{-1}$; Table 3) at this sampling interval suggested resuspension as more likely than *in situ* production given the absence of sea ice. During this period, there was also a large flux of chloroplast-containing *Cylindrotheca clos-terium*, a rapid growing diatom when resuspended in the euphotic zone and common on shallow shelves, among other diatoms that suggested a resuspension-driven autumn bloom as sea ice was absent and sunlight sufficient for growth [49]. In addition, water temperatures, salinity and nutrient data collected at the CEO as well as the meteorological record from the US National Weather Service station in Utqiaġvik indicated an increase in storm frequency and intensity during this period [68]. These fall storms generally lead to a mixing of the water column, bringing remineralized nutrients to the surface, and allowing for the possibility of an autumn bloom [69].

## Latitudinal gradients of sympagic HBIs and declining sea ice

While core tops can provide more reliable collection of undisturbed surface sediments, comparisons of surface sediments collected by Van Veen grabs and Haps core tops in this biologically productive region were found to have no significant difference in radiocesium activity, suggesting similar recent deposition [53]. Therefore, we are confident that the results from the surface sediment analysis present recent deposition with some degree of interannual variability, but unlikely to represent a single year due to the mixing on the shelf.

$IP_{25}$ and HBI II were detected throughout our study sites in the Bering and Chukchi Seas. The range of $IP_{25}$ concentrations in the surface sediments (0–12 µg $g^{-1}$ TOC), are comparable with the range of previously reported pan-Arctic observations (0–10 µg $g^{-1}$ TOC) [70]. The largest concentration observed (12 µg $g^{-1}$ TOC), in addition to samples with values exceeding 10 µg $g^{-1}$ TOC (n = 4), suggest there were localized areas of elevated ice algal export in the Pacific Arctic. One prior study of $IP_{25}$ in the Pacific Arctic indicated comparable concentrations (0–5 µg $g^{-1}$ TOC [37]). However, direct comparisons with our data may be equivocal because of the less productive location further offshore near the Chukchi Borderlands. In

addition, there were methodological differences in the prior study because instrument response factors were not taken into account in their $IP_{25}$ estimates.

When the H-Print index was compared with two satellite-derived sea ice metrics (mean spring sea ice concentrations and ice-free period before sample collection), there was general agreement regarding the periods of open water and sea ice cover for each season (Figs 5 and 6). The H-Print was a slightly better predictor of the mean spring sea ice concentration rather than break-up date, likely due to the scale of this measurement and the resolution of the satellite data. As was the case with the sediment trap analysis, the snowmelt period prior to break-up was the event that initiated the biomarker flux consistent with an ice algae bloom. This parameter likely signified melt pond formation and melting of the bottom few centimeters of sea ice. This represents an advantage over satellite-based observations that do not indicate whether there was significant production occurring beneath the ice.

H-Print indices from 2014–2017 show significant sea ice algal deposition, and increasing proportions of sympagic inputs on a latitudinal gradient (Figs 3 and 6A). Pelagic influences were significantly greater in the northern Bering Sea and southeast Chukchi Sea than in the northeast Chukchi Sea. However, individual biomarkers provide a more nuanced perspective of localized areas of elevated ice algae markers. Sea ice algal material deposition was increasingly significant throughout the northeast Chukchi shelf, southeast of Hanna Shoal and in upper Barrow Canyon (Fig 3).

**Northern Bering and Southeast Chukchi Seas.** Although the H-Print suggests proportionally low ice algae deposition throughout the 68–70˚N stations overall, there were occurrences of elevated $IP_{25}$ concentrations relative to all sampling locations. These localized areas were observed in the SLIP, CHIR, and SECS regions and contained some of the highest concentrations observed in this study. For example, in the SECS region in 2015, station SEC6 had an $IP_{25}$ concentration of 11 µg g$^{-1}$ TOC but an H-Print of 74%, suggesting greater pelagic influence. These cases in which there are high $IP_{25}$ concentrations with higher H-Print values (>50%) can be explained by a significant contribution in mass by sea ice algae but not necessarily the proportion of total production that may be sustained in the open water season by pelagic production [15]. This could also be attributed to environmental drivers, such as nutrient limitation increasing HBI production [65]. Given that the H-Print is determined as a ratio of the pelagic HBI to total HBIs, this index may reduce the prominence of the early season input of ice algae in the northern Bering Sea where phytoplankton blooms are substantial in the summer months and can also experience autumn blooms [71]. The apparent dominance of the pelagic signature is consistent with the longer open water period and more time for pelagic phytoplankton production compared to the study area to the north. However, there were a few notable exceptions to the high $IP_{25}$ coinciding with high H-Print scenarios. For example, at the SLIP3 station in 2015 we observed a low H-Print (35%) and high concentration of $IP_{25}$ (12 µg g$^{-1}$ TOC; S1 Table). This is the general location of the recurring St. Lawrence Island polynya that forms in the winter, enhancing the production of sea ice and late winter production, but these data suggest that summer open water production is not as prominent. HBI profiles in sediments near a polynya have not been widely described or reported, but this could be one explanation for this observation.

Advection through the Pacific Arctic region provides an important source of nutrients and organic matter. Upstream production of ice algae could be a contributing fraction of the material carrying the $IP_{25}$ observed in the sediment trap prior to ice melt in the northeast Chukchi Sea, given the pattern of sea ice retreat. The appearance of $IP_{25}$ in the surface sediments at these lower latitude stations does suggest the sinking of some portion of this production. However, retention of $IP_{25}$ is likely greater in SLIP and SECS than in CHIR based on larger sediment grain size [15] and stronger currents in the Chirikov Basin as the flow pathways

converge entering Bering Strait [42, 72]. There is generally limited pelagic grazing by zoo-plankton at the time of ice algal production in the SLIP region, allowing for the organic matter to settle largely unaltered to the benthos [8, 16, 73].

The SLIP region has been undergoing a shift in the arrival, retreat and duration of sea ice in the past several years [34, 64]. There was an unprecedented decrease in sea ice duration in this region in 2014/15, 2016/17 and 2017/18 [64]. H-Print values for surface sediments in the 2015–2017 seasons are consistent with these indications of open water productivity. If the current trend in the SLIP region towards more ice-free conditions year round continues, early ice algal production will be increasingly removed from the local food web; water column stratification may not occur until later in the season, which could result in decreased phytoplankton production [64].

**Northeast Chukchi Sea.** Among the biomarkers studied here, sympagic HBIs were the dominant contributor in the NECS for all years sampled. Given the insights from data on the ice algal fluxes at the CEO, it is reasonable that ice algal production, export, advection, and resuspension sustain a year round source of sea ice algal material to the benthos of the Chukchi Shelf. However, particulate organic carbon and diatom export have been found to be highly variable on the Chukchi shelf [74]. Surface sediments collected at stations nearest the moored CEO sediment trap show some of the highest concentrations of $IP_{25}$ and HBI II observed in this study. In addition, *N. frigida* and *M. arctica* fluxes, which are generally low on Arctic shelves, were higher at the CEO sediment trap in the northeast Chukchi Sea than fluxes observed in the Beaufort Sea and the Eurasian Arctic [11, 75], suggesting elevated sea ice algal export in 2016.

The NECS hotspot is known for high *in situ* production with pelagic and benthic retention in addition to the inputs of upstream productivity [15]. The flow is variable, paired with a heterogeneous bathymetry that promotes retention of cold and saline water that forms in the winter, carrying relatively high nutrient concentrations [40, 41]. Hanna Shoal is an important subsurface feature in the NECS, with active ice keeling and sea ice persistence after ice has melted elsewhere on the shelf [41]. Productivity is high along the southeastern flanks of Hanna Shoal, where strong pelagic-benthic coupling results in increased benthic biomass and foraging opportunities for walruses in the late summer [15, 16, 76, 77].

Barrow Canyon also appears to be a region of high ice algal material inputs due to the low H-Print values and low abundances of HBI III. Much of the current flow from the Chukchi shelf exits through Barrow Canyon, carrying organic matter towards the deeper Canada basin. Export fluxes of particulate matter are high both in the presence and absence of sea ice in Barrow Canyon with more labile, fresh organic matter exported than in other regions of the Chukchi shelf [74]. It is probable that there is local production of sea ice algae, given the dominance of sympagic HBIs, but sediments also contain advected material from the shelf. Consequently, sea ice algal material appears to make a significant contribution to the benthos at this study location in addition to also likely forming a source of sympagic production that is exported into the deeper basin.

## Sympagic HBI burial through bioturbation and sedimentation

The H-Print levels from the sediment core collected on the slope (NNE-14) were dominated by sea ice carbon biomarkers throughout the entire 20 cm core depth (Fig 9). The location of this core, near the median minimum limit of summer sea ice extent (1981–2010, Fig 2A), means it is likely representative of late season export and a shorter duration of open water relative to the shelf. Sedimentation rates for this core based on estimates from peak $^{137}$Cs activity (0.04 cm yr$^{-1}$) were similar to the estimate from $^{210}$Pb (0.02 cm yr$^{-1}$, data not shown),

suggesting a core spanning centuries of deposition. Based on radiocesium measurements throughout the shelf region, maximum $^{137}$Cs activity occurs between 6–10 cm depth, suggesting the surface sediments represent years and not decades or centuries of deposition [52]. While core tops can provide more reliable collection of undisturbed surface sediments, comparisons of surface sediments collected by Van Veen grabs and Haps core tops in this biologically productive region were found to have no significant difference in radiocesium activity, suggesting similar recent deposition [53]. The H-Print values were slightly higher in the top 6 cm (>20%, Fig 9), where the sediment characteristics were similar to the shelf, although still predominantly sympagic, suggesting a possible recent increase in pelagic phytoplankton deposition. In the bottom 8–20 cm of the core, where the composition consisted of grey, fine-grained sediments, the H-Print is relatively homogenous and strongly sympagic (8–20%, Fig 9). By comparison to the slope, unambiguous sedimentation rates cannot typically be estimated from cores collected on the Chukchi shelf due to the high degree of bioturbation [13, 52, 78]. The $^{137}$Cs profile from the station UTX 13–23 on the shelf (Fig 9) suggests a well-mixed profile and a somewhat mixed composition of HBIs at nearby DBO4.6 (H-Prints between 40 and 60%). However, there is an increasingly sympagic signature (~30%) at the bottom of the core, suggesting a persistence of the sympagic sourced organic matter at depth or possibly a reduction of sympagic production associated with sea ice declines. Since the shelf has higher nutrient loads, levels of productivity [14], and an earlier retreat of sea ice, it is not surprising the core collected at DBO4.6 indicates a greater influence of pelagic production than NNE14. The H-print data from NNE14 also reflects the limits of phytoplankton deposition relative to sea ice algal deposition on the slope, since this core was collected from a slope area that was historically close to the minimum extent of the ice edge or is ice-covered for most of the year.

The sediment core H-Print data collected near DBO4.6 supports the assumption that there is rapid burial of sea ice algae relative to phytoplankton. The propensity of ice algae to form aggregates, facilitated by microbial exopolymeric substances and the rapid sinking of the pennate diatom *N. frigida*, may indicate greater relative pulses of ice algae to the seafloor despite a larger relative proportion of pelagic productivity [79, 80]. These processes have also been suggested to support the greater burial potential of sympagic lipid biomarkers [66, 81]. The H-Print values also suggest there is a greater source of ice algae lipids available to the benthic infaunal communities that occupy these sediment horizons. HBI burial data are not available for cores spanning the entire shelf, but it can be expected from the surface sediment data presented in this study that it is likely that ice algal lipids are stored in sediments throughout the Bering and Chukchi shelf. The persistence and potential availability of labile ice algal lipids mixed to depth in the sediments is an important consideration for assessing the ecosystem response to the loss of seasonal sea ice. It is important to note that despite the high degree of bioturbation, the preservation of these biomarkers is still robust. IP$_{25}$ in particular has proven to be controlled more by climatic conditions rather than degradation processes [82]. According to Rontani et al. [83], autoxidation of lipids in the oxic layers of sediments can be particularly important in regions of low accumulation rates, where near-surface sediments can represent decades to centuries of deposition. There is relatively high deposition based on $^{137}$Cs sediment profiles throughout the Chukchi Shelf, where the $^{137}$Cs maxima associated with peak bomb fallout deposition (1963) averaged 7–8 cm in depth. Radiocesium based sedimentation estimates determined from these previous studies on the shelf ranged from 0.1 up to 0.3 cm yr$^{-1}$ [52], suggesting deposition on the scale of years in near-surface sediments.

## Mechanisms for HBI distribution throughout the Pacific Arctic

The gradient of HBIs throughout the northern Bering and Chukchi Sea sampling locations and the seasonal succession of sympagic to pelagic diatoms as determined through export fluxes at the CEO [49], suggests a general regionally-specific HBI production mechanism (Fig 10). In similarity to the use of HBIs in the Antarctic MIZ [67], the HBI distribution in the Pacific Arctic may be a proxy for relative sea ice persistence rather than proportions of production of sea ice algae and phytoplankton organic matter. In the more southerly latitudes of the northern Bering Sea (62–65°N), sea ice persistence typically occurs 0–3 months of the year. Sea ice retreat historically initiated early in the year (March-April), allowing for a spring sea ice algae bloom. The ice algae bloom is thought to seed a phytoplankton bloom as the ice retreats, with a gradual transition of sympagic to pelagic assemblages [8]. The more recent extended open water period in the northern Bering Sea region and a deepening of the mixed layer allows for a second phytoplankton bloom in the fall before sea ice freeze up, which may be particularly relevant during warmer years [84]. Therefore, sympagic HBI (IP$_{25}$ and HBI II) production likely occurs during the brief period in early spring with two possible pulses of HBI III production throughout the late spring and fall. This results in a greater relative proportion of the apparent pelagic HBIs relative to the sympagic-origin HBIs. There are also likely to be years with no new IP$_{25}$ or HBI II production due to the timing of sea ice retreat or lack of formation. The current flow over the Bering shelf, through Bering Strait and into the Chukchi shelf promotes the advection of HBIs northward, potentially elevating the HBI III proportionally in the southeast Chukchi Sea as currents slow north of the Strait. HBI flux data in the northern Bering Sea do not yet exist but could help to refine some of these assumptions.

In the northeast Chukchi Sea, sea ice coverage extends into the summer months (July-August), with some regions of localized persistence throughout the summer, particularly near Hanna Shoal [41]. Sea ice persistence at these higher latitudes typically occurs for 6–9 month intervals. Advection of HBIs from more southerly locations is likely but ultimately may be a minimal source deposited to the northern shelf sediments, due to the aggregation and rapid sinking of diatoms closer to the point of production [9]. The sympagic production initiates with increasing PAR followed by the release of ice algae in April-May, and an under-ice bloom composed of sympagic and pelagic diatoms from June to August as open water is initiated (Fig 10). The presence of exclusively pelagic diatoms reflected the development of an under-ice bloom, as observed in June and July 2016 [49], with HBI III export that coincides with pelagic-sourced production. However, the peak export of HBI III should occur after ice break up during the open water period. In this study, IP$_{25}$ export occurs year-round through both new production and resuspension. The appearance of *M. arctica* resting spores following the ice algae bloom and through the fall months supports the prevalence of sympagic diatom persistence in a sediment "seed bank", which can be resuspended in the fall [85]. The presence of IP$_{25}$ throughout the year may suggest that *Haslea* and *Pleurosigma* resting cells persist until the return of sea ice on the Chukchi shelf. Supporting evidence of this was observed in laboratory cultures of *H. crucigeroides* and *H. vitrea* maintained in complete darkness for over six months, where the cells remained viable and with their HBI content the same as when grown in light (unpublished data). Owing to the shallower conditions on the Chukchi shelf (40–50 m), it seems clear that resuspension during the open-water period plays an important role in the persistent IP$_{25}$ signal.

## Conclusions

Based on the results of this study, sea ice algae (or some component of sea ice algal origin i.e. lipids, fatty acids, hydrocarbons) are present year-round in the northeast Chukchi Sea with export events occurring to some degree at all phases of the sea ice cycle, along with seasonal

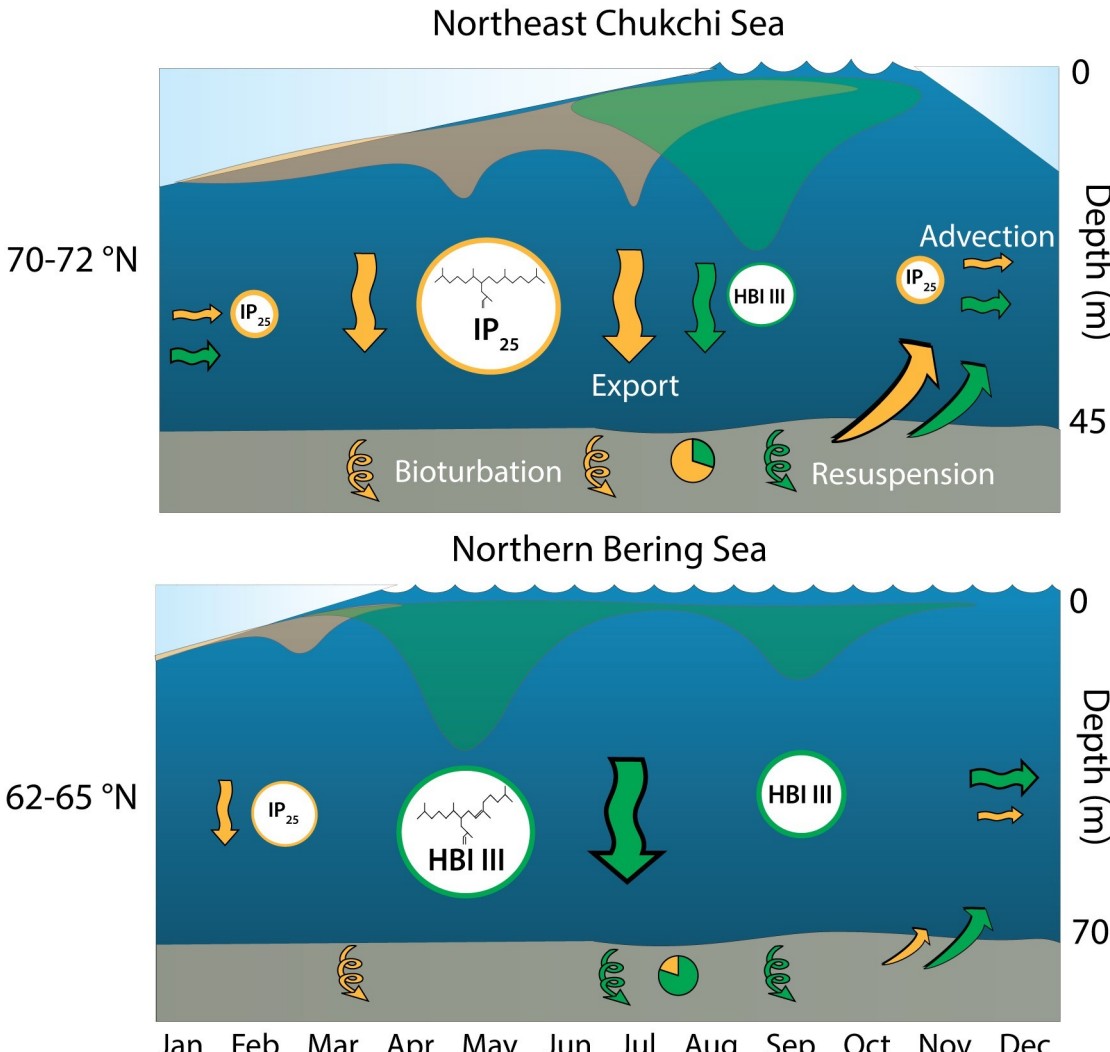

**Fig 10. Conceptual diagram for the production, flux and fate of HBIs in the Pacific Arctic.** Sea ice persistence increases from the northern Bering Sea to the northeast Chukchi Sea. There is a brief opportunity for sympagic production (yellow shading) in the Bering Sea due to the timing of sea ice retreat and return of sunlight, followed by extensive ice-edge and open water phytoplankton blooms (green shading) in the spring and fall. Sympagic production can occur over a longer period in the Chukchi Sea. Sympagic $IP_{25}$ production (yellow circles) occurs in much lower proportions to pelagic HBI III (green circles) owing to the extensive open water period in the northern Bering Sea. In the Chukchi Sea, there is a greater proportion of $IP_{25}$ to HBI III. This relative proportionality is observed in the surface sediments when sampled in the summer (pie chart). There is rapid burial of the sympagic HBIs (yellow spiral) owing to aggregation and rapid sedimentation in both regions, with a greater proportion available on the Chukchi shelf. Resuspension (upward arrows) plays a larger role in the Chukchi Sea, sustaining the suspension of $IP_{25}$ and in the water column. Advection (horizontal arrows) is also likely to be a more prominent contribution to the HBI signal in the Chukchi than the northern Bering Sea. Symbols courtesy of the Integration and Application Network, University of Maryland Center for Environmental Science (ian.umces.edu/symbols/) and reprinted under a CC BY license, permission from B. Walsh, original copyright 2020.

resuspension events. This study also confirms that satellite observations underestimate the ice algal component due to peak export occurring during snow melt that happens before sea ice break up. The presence of $IP_{25}$ without strong indications of the associated diatoms present emphasizes the need for future investigations on $IP_{25}$ synthesis using ice cores from the Bering and Chukchi seas and the possibility of identifying other species that are capable of producing these compounds. Given the overlap of HBI III production with *Pleurosigma* spp., the weaker

correlations with *Rhizosolenia* spp., and correlations with sympagic HBIs, the need to determine the fidelity of truly pelagic HBI biomarkers is still an ongoing imperative.

This study presents an assessment of the production, flux and fate of HBI biomarkers using the H-Print sea ice index in the Bering-Chukchi Sea inflow shelf system. We found evidence of a northward latitudinal gradient of decreasing pelagic to sympagic production proportionality in the Pacific Arctic system likely driven by sea ice persistence. These data indicate that sea ice algae contribute a significant portion of the organic matter deposited to the seafloor in the NE Chukchi Sea, with a peak early spring pulse and year-round persistence. With a foundational understanding and baseline measurements of the production and distribution mechanisms of HBIs in the Pacific Arctic region, these lipid biomarkers may serve as an integrating tool to better understand and monitor the rapid changes occurring in this ecosystem, which are associated with shifts in the timing and distributions of primary production with cascading effects in the food web. HBIs provide a targeted approach to isolating the sea ice algae contributions that other methods lack (e.g. stable isotopes, fatty acids). However, there are still limitations as these biomarkers are proxies and may not always faithfully reflect the community composition. Setting the region apart from the rest of the Arctic, the Pacific Arctic is one of the world's most productive ocean ecosystems [86] with nutrient-rich waters allowing for high primary production, emphasizing the importance of regional considerations when applying HBI biomarkers to paleoclimate studies. This includes the influence of physical drivers, nutrient dynamics, primary production rates and phytoplankton community composition that likely influence the abundance and proportion of HBI production.

## Supporting information

**S1 Table. Surface sediment sample summary.** Summary of surface sediment sample station names and coordinates (latitude/longitude), dates and cruises collected, TOC (%), and HBI biomarker concentrations including $IP_{25}$ (μg/g TOC), HBI II (μg/g TOC) and HBI III (μg/g TOC) along with H-Print (%) values.
(XLSX)

## Acknowledgments

We thank the science teams, captains and crew aboard the CCGS *Sir Wilfrid Laurier* and USCGC *Healy*. We also thank Carla Ruiz-Gonzalez (Scottish Association for Marine Science) for assistance with sample processing and Laura Gemery (United States Geological Survey) for access to archived frozen sediment samples. We would also like to thank Cheryl Clark and Andrew Heyes for the Organic Analytical Laboratory facilities and support at the Chesapeake Biological Laboratory. We thank two anonymous reviewers for critical comments that improved an earlier version of the manuscript and R. Schlitzer for providing permission to use Ocean Data View images for publication in PLoS One under the Creative Commons Attribution License (CCAL) CC BY 4.0.

## Author Contributions

**Conceptualization:** Chelsea Wegner Koch.

**Formal analysis:** Chelsea Wegner Koch.

**Funding acquisition:** Lee W. Cooper, Jacqueline M. Grebmeier.

**Investigation:** Chelsea Wegner Koch, Lee W. Cooper, Catherine Lalande, Thomas A. Brown, Karen E. Frey, Jacqueline M. Grebmeier.

**Methodology:** Catherine Lalande, Thomas A. Brown.

**Supervision:** Lee W. Cooper.

**Visualization:** Chelsea Wegner Koch.

**Writing – original draft:** Chelsea Wegner Koch.

**Writing – review & editing:** Chelsea Wegner Koch, Lee W. Cooper, Catherine Lalande, Thomas A. Brown, Karen E. Frey, Jacqueline M. Grebmeier.

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
