## [Decision Letter · Decision Letter 0]

12 Feb 2020

PONE-D-19-34317

Seasonal succession and latitudinal gradients of sea ice algae in the Northern Bering and Chukchi Seas determined by algal biomarkers

PLOS ONE

Dear Chelsea Koch,

Thank you for submitting your manuscript to PLOS ONE. After careful consideration, we feel that it has merit but does not fully meet PLOS ONE’s publication criteria as it currently stands. Therefore, we invite you to submit a revised version of the manuscript that addresses the points raised during the review process.

Based on my own evaluation and that of two independent reviewers, I believe that this study provides an important contribution to our understanding of sea ice algae and associated biomarkers in this region. I look forward to receiving and reading a revised submission based on the detailed reviewer comments below.

We would appreciate receiving your revised manuscript by Mar 28 2020 11:59PM. To enhance the reproducibility of your results, we recommend that if applicable you deposit your laboratory protocols in protocols.io, where a protocol can be assigned its own identifier (DOI) such that it can be cited independently in the future. For instructions see: http://journals.plos.org/plosone/s/submission-guidelines#loc-laboratory-protocols

We look forward to receiving your revised manuscript.

Kind regards,

Christof Pearce

Academic Editor

PLOS ONE

Journal Requirements:

3. We note that Figures 2, 4 and 5 in your submission contain map images which may be copyrighted. All PLOS content is published under the Creative Commons Attribution License (CC BY 4.0), which means that the manuscript, images, and Supporting Information files will be freely available online, and any third party is permitted to access, download, copy, distribute, and use these materials in any way, even commercially, with proper attribution. For these reasons, we cannot publish previously copyrighted maps or satellite images created using proprietary data, such as Google software (Google Maps, Street View, and Earth). For more information, see our copyright guidelines: http://journals.plos.org/plosone/s/licenses-and-copyright.

You may seek permission from the original copyright holder of Figures 2, 4 and 5 to publish the content specifically under the CC BY 4.0 license. 

If you are unable to obtain permission from the original copyright holder to publish these figures under the CC BY 4.0 license or if the copyright holder’s requirements are incompatible with the CC BY 4.0 license, please either i) remove the figure or ii) supply a replacement figure that complies with the CC BY 4.0 license. Please check copyright information on all replacement figures and update the figure caption with source information. If applicable, please specify in the figure caption text when a figure is similar but not identical to the original image and is therefore for illustrative purposes only.

Reviewers' comments:

Reviewer's Responses to Questions

**Comments to the Author**

1. Is the manuscript technically sound, and do the data support the conclusions?

Reviewer #1: Partly

Reviewer #2: Yes

2. Has the statistical analysis been performed appropriately and rigorously? 

Reviewer #1: Yes

Reviewer #2: Yes

3. Have the authors made all data underlying the findings in their manuscript fully available?

Reviewer #1: Yes

Reviewer #2: Yes

4. Is the manuscript presented in an intelligible fashion and written in standard English?

Reviewer #1: Yes

Reviewer #2: Yes

5. Review Comments to the Author

Reviewer #1: This manuscript presents data on the production and fate of sea ice and pelagic biomarkers in the Bering-Chukchi Seas region. The authors make use of surface sediment, sediment trap, and sediment core samples collected at different times in the region and attempt at assessing the fate of these biomarkers in a conceptualized manner, mainly based on the H-print index.

The main strengths of this work are, in my opinion, 1) the fact that it covers a large latitudinal gradient and presents data from a still poorly-known region of the Arctic; 2) the fact that it includes data from surface sediments, traps and cores and 3) the fact that it highlights the importance of snow melt rather than sea ice melt in triggering the under-ice bloom, which again confirms the limitations of satellite observations for predicting ecosystem dynamics.

While the study includes interesting data, it unfortunately lacks a clear focus or hypothesis and has some methodological shortcomings perhaps due to the unclear goal. It appears that these data were collected for other purposes, and later on assembled together. I believe this material, if explored fully, had the potential to considerably advance our understanding of Arctic sea ice species and biomarker dynamics in time and space for this region. Instead, and mainly due to methodological limitations, the study does not quite provide new knowledge on sea ice-related diatoms per se, but it thus offers useful new data on how HBIs and the H-print capture sympagic/pelagic production along a latitudinal gradient in this region. I encourage the authors to define clearly the main purpose and finding(s) of this study and refocus the manuscript accordingly.

My main concerns are:

1) If the goal of the study was, as indicated by the title, to provide new insights into sea ice algae succession and latitudinal gradients, then the methods used are not adequate for this purpose. The authors have not looked at species succession but rather selected a priori indicators that have been grouped at rather low taxonomic resolution, so this study offers no actual information on diatom succession. It would indeed have been very interesting to see a record of diatom taxa from the trap, but this is not possible using the Utermohl method applied here. In order to be able to identify the diatom species present, it would have been necessary to do a diatom rinse of the trap sediments to see details of the frustules at high resolution, such as routinely done in micropaleontology. This would have circumvented some of the caveats associated with grouping together e.g. Gyrosigma/Pleurosigma/Haslea species. The full trap data published by Lalande et al should be better discussed and incorporated here. For example, it isn’t clear what was the relative abundance of the groups used here in relation to the 300 phytoplankton cells counted per sample.

2) In order to correctly assess fluxes of biomarkers from the trap samples, and compare these with the surface sediment data, the biomarker values should have been normalized by TOC. However, as far as I could see, TOC analyses were not performed on the trap samples. This limits the comparison between different trap samples, and this should be noted/discussed.

3) Sampling with a van veen grab is not ideal for collecting undisturbed surface sediments. Sedimentation rates across the study region are very variable, and this should be clearly mentioned and discussed. What time interval are the surface samples expected to cover?

4) 137Cs measurements do provide a first order idea of mixing, but in order to assess sedimentation rates in the cores and estimate their age, 210Pb analyses should have been done as well – I encourage the authors to measure 210Pb activity in these samples if there is available material from the cores. With only 137Cs available, the conclusions that can be drawn are rather limited. Also, the two cores, given their different settings, could allow for a more in-depth discussion of deposition vs. bioturbation and preservation of biomarkers e.g. it is expected that bioturbated sediments are more exposed to oxygen/degradation and this might be reflected in their biomarker record.

Detailed comments:

I suggest a different title, to better capture the essence of the study e.g. Temporal and spatial dynamics of sea ice-related biomarker production and deposition in the Northern Bering and Chukchi Seas

Line 46-47 – this is an outdated/oversimplified list. Include heterotrophic and mixotrophic protists. Bacteria are listed twice.

Lines 74-75: later on in the text it is mentioned that HBI III is also an indicator of MIZ. To avoid confusion, this should be mentioned here as well.

Fig.1 – Is this figure justified/necessary?

Lines 93-97 – The justification for the study is rather vague. There seem to be two overall motivations: 1) lack of data from the region; 2) understanding dynamics of these biomarkers in order to better apply them to ecosystem and paleoclimate studies. I suggest sharpening this part and clearly stating the goal(s) of the study. And then truly discussing this in the end. As it stands, the discussion only briefly mentions implications of the results for ecosystem studies, but not for paleoclimate studies.

Lines 119-127 – I am not aware of any studies testing the possible effects of formalin and preservation of sediment trap samples on HBIs. Were the trap samples kept cold after recovery? Please provide details.

Line 141 – Limoges et al 2018 indicate that H. spicula, not H. crucigeroides is an IP25 producer. In the Brown et al study, the authors did not distinguish between these two species. Add reference.

Lines 142-144 – As mentioned earlier, the limitation is not the use of microscopy per se. It is the use of fresh samples/Utermohl method. If instead, cleaned frustules were examined at 1000X resolution with phase contrast it would have been possible to identify most of the species present.

Fig. 2 – sediment sampling locations “were selected” instead of “occurred”.

Lines 295 – Do you mean the opposite – i.e. concentrations decreased in late July?

Lines 316-317 – Here it would be good to see fluxes normalized by TOC. Could the apparent decline in IP25 actually reflect an increase in total flux rates?

Fig. 3 – It is important to add to the figure legend and the figure itself what year(s) the trap data cover (2015-2016). I don’t understand what “pieces” stands for – fragments? If so, why not include these in the spp. counts? And how large fragments were considered =1?

Figs. 4 and 5 Legend – it should be clearly stated in the legend that sampling stations are not the same for each year. Sampling sites are not easy to see as they are plotted, and at a quick glance the figure could be mis-interpreted as showing a southward expansion of HBIs over time.

Lines 668-670 – Is this supported by the data?

Lines 680-681 – explain what evidence you have to support this – not clear what data are behind the assumption that this is “a likely but minimal source”

Lines 688-690 – resting spores are dense and primed for sinking into the seafloor. It is more likely that the blooms are seeded from sediments than from the water column. Unless the life-cycle of M arctica is well studied, it is speculative to assume they survive in the water column.

Lines 690-691 – this statement seems rather odd. Of course sea ice species have strategies to persist when sea ice is not present, as do all other aquatic protists. Dormancy is a wide-spread strategy in protists. Ellegaard and Ribeiro 2018 review the phenomenon of long-term dormancy of microalgae in aquatic “seed banks” – article published in Biological Reviews.

Lines 692-694 – very interesting finding – what species were kept in the laboratory? As T. Brown is also an author here, I suppose you could refer to it as “our own unpublished results” rather than a personal comm.

Conclusions – the entire first paragraph of the conclusions should be moved/merged into the discussion section.

I encourage the authors to consider what are the implications of their findings – back to the stated goals at the end of the introduction. For example, why and how may “lipid biomarkers serve as an integrating tool to better understand and monitor the rapid changes occurring in this ecosystem”? What are the potentials and limitations? And what issues need to be taken into account that are specific to this region, but perhaps do not apply to other parts of the Arctic?

Reviewer #2: This paper presents highly branched isoprenoid (HBI) biomarker, including IP25, data and diatom data from a sediment trap in the Chukchi Sea and a suite of surface sediments across the Chukchi and Bering seas. It is among the first, perhaps the first, to report such data. This data is used to address the question of how much primary productivity occurs in ice covered waters. The authors propose a model for sea ice and pelagic diatom productivity and deposition across this highly productive region. It is an important paper that nicely summarizes the research and understanding of phytoplankton and sympagic algae.

I can’t comment on the biomarker/HBI methods, but I hope another reviewer was asked to look at this paper who is an HBI expert. I know that these methods have been tricky for some to properly emulate.

This is a well-written paper. The discussion is a bit long and could perhaps be shortened by editing and reorganizing the content that is currently on pages 22-25. But, I don’t have any significant comments or concerns. Some care needs to be given to the figures, which are quite pixelated in the pdf version of the manuscript. Several need additional annotations or the figure caption doesn’t match what is shown in the figure. Minor line by line, and figure by figure comments are below.

Minor line by line comments:

Line 100: Refer to Fig. 2

Line 125: I was struck by how high the salinity was adjusted to in the collection cups. Is there a reason for this high salinity?

Lines 190-197: An additional sentence describing how you expect the 137-Cs profile to be similar to DBO 4.6 or why a core 50 nm away is expected to be an adequate substitution would be helpful.

Line 296: I’m not sure why you say, “sea ice concentration never dropped below 15% before the end of the sediment trap deployment.” From figure 3, it looks like sea ice drops below 15% in July, peaks just above 15% for a brief moment and then is below 15% when the trap is recovered.

Line 322: It would be nice to remind the reader here that high H-print values indicate high pelagic contributions.

Line 337: The H-print is really the proportion of pelagic to sympagic, not the other way around. Also, it indicates higher contributions of pelagic diatoms, not necessarily greater periods of ice free waters. It might be a proxy for ice, but it’s really just measuring diatom contributions.

Lines 379-380: I suggest removing the words “and sea ice cover” and “greater periods of ice-free surface waters” because H-print really indicates the algal contribution not actually sea ice.

Line 438: What does “clayish” mean? Clay is a textural term meaning grains smaller than 4 um (or < 2 um if you’re a soil scientist). Do you mean clay-rich? Silt and clay? Fine grained?

Lines 468-479: I think the authors overstate the need to be cautious here. Although I agree, that there is a reason to be cautious with IP25 because the proxy is really based on species that arguably are very minor, the results that are presented actually strengthen the interpretation that IP25 is an appropriate proxy for sea ice. I would suggest replacing the clause, “suggests the need for further studies before a final interpretation can be made.” with the opposite, “strengthens our interpretation.”

Line 499: “A coeval of HBI II and HIB III” is strange wording. Perhaps you mean, “An association between HBI II and HBI III”?

Line 572: Since we don’t know why diatoms produce IP25, we don’t really know whether in places where there is an increase in IP25 if it’s an increase in the number/mass of those diatoms, or just some kind of environmental event that causes the diatoms to produce IP25. It’s probably a good idea to keep this in mind.

Line 589: Please remind the reader whether you’re referring to larger or smaller grain size.

Line 597: I think you’re missing the word, “continues” between, “year round,” and, “early ice.”

Line 636: I think you mean the 137Cs profile from core UTX 13-23, not DBO4.6.

Line 639-657: Although DBO4.6 has more bioturbation than NNE14, it likely still gets older as you increase in depth. There is slightly less 137-Cs at depth, and I suspect that if you were to core deeper, you’d see the loss of 137-Cs and reach quite old sediments. In strongly bioturbated regions, the 137-Cs peak is smeared, but not necessarily obliterated. I think that the increased sympagic signature at depth is also likely due to decreasing sea ice over the past few decades.

Figures:

Figure 2: It would be helpful to label the boxes SLIP, CHIR, SECS, NECS, and BARC, and also repeat the boxes on Figure 4.

Figures 4 and 5: It’s really difficult to see the stations on these figures. Maybe make them a hair larger and colored solid black instead of grey. Also, it would be helpful to reproduce the boxes around the different regions (maybe remind us in the figure caption their names north to south?) since you refer to the regions in the text.

Figure 6: You reversed A and B in the figure/caption.

Figure 7: The dots need to be slightly bigger on these plots. It’s impossible to distinguish colors, especially on panel B, but also the boxes in the legend on panel A are very small. For example, I can’t tell the difference between the color for NECS and SECS. Maybe you could just label the box plot?

Figure 10: I love this figure, but it’s impossible to read the text/labels in the molecular diagrams. In your figure caption, you label IP25 as red and HBI III as blue, but it appears to be the opposite in the figure. I’m also not sure what the scratch marks are on the underside of the ice in the Chukchi Sea Nov-Dec. Please describe what the brown and green shading indicates also (sea ice vs. pelagic diatoms?).

Table 1: It would be helpful to include the distance from the CEO sediment trap for each core location in this table. I’m also a little confused with the table and figure captions embedded in the text. Is line 187 part of the table caption? If so, then that’s fine. If not, you already said this earlier in the text.

6. PLOS authors have the option to publish the peer review history of their article (what does this mean?). If published, this will include your full peer review and any attached files.

Reviewer #1: No

Reviewer #2: No

---

## [Author Response · Author response to Decision Letter 0]

6 Mar 2020

Author Responses to Journal Requirements: 

We have revised the symbol to denote the corresponding author on the first page. Additionally, minor adjustments were made throughout the manuscript (e.g. sentence cases on section titles, spacing on table legends) to ensure that all style requirements have been met. 

We have added a section titled “Permitting” to the Methods outlining why we did not require permits.

3. We note that Figures 2, 4 and 5 in your submission contain map images which may be copyrighted. All PLOS content is published under the Creative Commons Attribution License (CC BY 4.0), which means that the manuscript, images, and Supporting Information files will be freely available online, and any third party is permitted to access, download, copy, distribute, and use these materials in any way, even commercially, with proper attribution. For these reasons, we cannot publish previously copyrighted maps or satellite images created using proprietary data, such as Google software (Google Maps, Street View, and Earth). For more information, see our copyright guidelines: http://journals.plos.org/plosone/s/licenses-and-copyright.

1. You may seek permission from the original copyright holder of Figures 2, 4 and 5 to publish the content specifically under the CC BY 4.0 license. 

We have received written permission from Reiner Schlitzer from Ocean Data View (ODV) to print the maps made with the ODV software. The email confirmation has been submitted as an “Other” file. The figure captions for Figs 2, 4 and 5 include the requested language for permission to print under a CC BY 4.0 license. 

In an additional email from PLOS, we were instructed to also complete this for Figure 10. This figure was made in Adobe Illustrator by modifying several symbols (in addition to original artwork) available through the University of Maryland’s Integration Application Network (IAN) symbol library for Adobe. IAN only requires the following: 

Required Attribution: Symbols courtesy of the Integration and Application Network, University of Maryland Center for Environmental Science (ian.umces.edu/symbols/). 

To ensure this is covered, we have also asked that they complete the form to publish under a CC BY 4.0 license. This signed form has been included. The Figure 10 caption includes IAN’s required attribution but also the statement recommended by PLOS for the CC BY 4.0 license. If this specific permission was not required for this figure, we can remove this text. 

Author Responses to Reviewer Comments

Reviewer #1: This manuscript presents data on the production and fate of sea ice and pelagic biomarkers in the Bering-Chukchi Seas region. The authors make use of surface sediment, sediment trap, and sediment core samples collected at different times in the region and attempt at assessing the fate of these biomarkers in a conceptualized manner, mainly based on the H-print index.

The main strengths of this work are, in my opinion, 1) the fact that it covers a large latitudinal gradient and presents data from a still poorly-known region of the Arctic; 2) the fact that it includes data from surface sediments, traps and cores and 3) the fact that it highlights the importance of snow melt rather than sea ice melt in triggering the under-ice bloom, which again confirms the limitations of satellite observations for predicting ecosystem dynamics.

While the study includes interesting data, it unfortunately lacks a clear focus or hypothesis and has some methodological shortcomings perhaps due to the unclear goal. It appears that these data were collected for other purposes, and later on assembled together. I believe this material, if explored fully, had the potential to considerably advance our understanding of Arctic sea ice species and biomarker dynamics in time and space for this region. Instead, and mainly due to methodological limitations, the study does not quite provide new knowledge on sea ice-related diatoms per se, but it thus offers useful new data on how HBIs and the H-print capture sympagic/pelagic production along a latitudinal gradient in this region. I encourage the authors to define clearly the main purpose and finding(s) of this study and refocus the manuscript accordingly.

We thank the reviewer for pointing out that additional efforts were needed to better define the goals and findings of the study. We have added a statement to more clearly identify the main purpose of the study in a new last paragraph at the end of the introduction. We agree that some additional knowledge may be potentially available from additional study of the Arctic sea ice species and biomarker dynamics. Plankton samples collected as part of this study are being taxonomically identified on an annual basis and will be reported separately as part of the overall Distributed Biological Observatory program. While the reviewer’s points are good with respect to better identification processes for ice algae and phytoplankton collected, the time, effort and resources required made this impractical to include within the scope of the current study. 

My main concerns are:

1) If the goal of the study was, as indicated by the title, to provide new insights into sea ice algae succession and latitudinal gradients, then the methods used are not adequate for this purpose. The authors have not looked at species succession but rather selected a priori indicators that have been grouped at rather low taxonomic resolution, so this study offers no actual information on diatom succession. It would indeed have been very interesting to see a record of diatom taxa from the trap, but this is not possible using the Utermohl method applied here. In order to be able to identify the diatom species present, it would have been necessary to do a diatom rinse of the trap sediments to see details of the frustules at high resolution, such as routinely done in micropaleontology. This would have circumvented some of the caveats associated with grouping together e.g. Gyrosigma/Pleurosigma/Haslea species. The full trap data published by Lalande et al should be better discussed and incorporated here. For example, it isn’t clear what was the relative abundance of the groups used here in relation to the 300 phytoplankton cells counted per sample.

We thank the reviewer for pointing out the limited scope of the taxonomic analysis. We have revised our statement in the discussion where we acknowledge the limitations of the methods used. Because our analyses are not as detailed as would be required for micropaleontological applications, we have removed the use of the word “succession” throughout the paper and have modified where possible, additional discussion from Lalande et al. 2020, where diatom fluxes of the most abundant taxa were analyzed. We also wanted to avoid duplicating information in Lalande et al. 2020. 

2) In order to correctly assess fluxes of biomarkers from the trap samples, and compare these with the surface sediment data, the biomarker values should have been normalized by TOC. However, as far as I could see, TOC analyses were not performed on the trap samples. This limits the comparison between different trap samples, and this should be noted/discussed.

We do have available POC flux data (published in Lalande et al. 2020), which can provide a sense of the overall flux but normalizing the HBI fluxes by TOC (or POC) is actually not a standard practice. There are a few existing studies reporting HBI fluxes, but none have presented HBI data normalized by TOC, e.g. Bai et al. 2019, Lalande et al. 2016, Fahl and Stein 2012. Belt 2018 provides a review furthermore of the latest protocols for HBI analysis and have been used as guiding principles. Normalizing by TOC is common practice, however, for sediment analysis which has been done here. By normalizing the HBI data, our results would not be comparable to previous studies. However, the reviewer raises an interesting point that should be discussed among the HBI community to establish best practices and determine whether this should be done. We have added the POC data 

3) Sampling with a van veen grab is not ideal for collecting undisturbed surface sediments. Sedimentation rates across the study region are very variable, and this should be clearly mentioned and discussed. What time interval are the surface samples expected to cover?

We have modified the methods to make it clear that the surface sediments were removed from the top of the grab through a trap door before the grab was opened. While clearly not as undisturbed as a coring device, a prior study (Cooper et al. 1998), which is now referenced in the text, showed that bioturbation is significant enough on the shelves of both the Bering and Chukchi seas that surface sediments collected in this way by both grabs and corers are not significantly different in bomb fallout activities from each other, indicating that both are affected by significant bioturbation. 

Additionally, H-prints were compared between Haps core tops and Van Veen samples at 4 locations. The largest difference in H-print was 6%, which is within the margin of error (12%) and now reported in the manuscript.

4) 137Cs measurements do provide a first order idea of mixing, but in order to assess sedimentation rates in the cores and estimate their age, 210Pb analyses should have been done as well – I encourage the authors to measure 210Pb activity in these samples if there is available material from the cores. With only 137Cs available, the conclusions that can be drawn are rather limited. Also, the two cores, given their different settings, could allow for a more in-depth discussion of deposition vs. bioturbation and preservation of biomarkers e.g. it is expected that bioturbated sediments are more exposed to oxygen/degradation and this might be reflected in their biomarker record.

While we determined radionuclide distributions in two cores, we concluded that determining sedimentation rates or age models for these cores was outside of the scope of this study, which was focused on biomarker distributions specifically. Data in Cooper and Grebmeier (2018), as well as other studies that are referenced in the manuscript provide much more context for sedimentation processes on the Chukchi shelf. For example, 210Pb data for the slope core (NNE-14) are available, and the sedimentation rate determined is now mentioned in the text (fifth line in section “Sympagic HBI burial through bioturbation and sedimentation”, but we chose not to formally present the data because it does not add that much to the story, particularly when a much larger set of core data over a much larger area of the Chukchi shelf are available in Cooper and Grebmeier (2018). These data from core NNE-14 will be presented in the first author’s PhD dissertation. 

We also addressed the concerns of abiotic and biotic degradation of HBIs in bioturbated sediments, primarily referencing Rontani et al (2018 and 2019) work. According to Rontani et al. 2019, autoxidation of lipids in the oxic layers of sediments can be particularly important in regions of low accumulation rates, where near-surface sediments can represent decades to centuries of deposition. Based on the sedimentation rates on the Chukchi Shelf, there is high deposition, where 137Cs peaks (if prominent and not mixed) often reached 7-8 cm in depth. Sedimentation rates that were determined ranged from 0.1 up to 0.3 cm/yr. This suggests the surface deposition likely represents years rather than decades or centuries. 

Detailed comments:

I suggest a different title, to better capture the essence of the study e.g. Temporal and spatial dynamics of sea ice-related biomarker production and deposition in the Northern Bering and Chukchi Seas 

Revised

Line 46-47 – this is an outdated/oversimplified list. Include heterotrophic and mixotrophic protists. Bacteria are listed twice.

Revised

Lines 74-75: later on in the text it is mentioned that HBI III is also an indicator of MIZ. To avoid confusion, this should be mentioned here as well.

Revised

Fig.1 – Is this figure justified/necessary?

We think the figure is helpful because readers will include not only those using HBI indices but also researchers working in the Pacific Arctic region on a number of other pressing scientific challenges involving sea ice retreat. Providing some information for readers unfamiliar with the methodology and approach will stimulate interest in the addition of HBI measurements to future studies. Additionally, the chromatograms are specific examples from our instrumentation that demonstrate accurate identification of the biomarkers. 

Lines 93-97 – The justification for the study is rather vague. There seem to be two overall motivations: 1) lack of data from the region; 2) understanding dynamics of these biomarkers in order to better apply them to ecosystem and paleoclimate studies. I suggest sharpening this part and clearly stating the goal(s) of the study. And then truly discussing this in the end. As it stands, the discussion only briefly mentions implications of the results for ecosystem studies, but not for paleoclimate studies.

Revised the overall motivations and goals, particularly at the end of the introduction. 

Lines 119-127 – I am not aware of any studies testing the possible effects of formalin and preservation of sediment trap samples on HBIs. Were the trap samples kept cold after recovery? Please provide details.

Details are now provided. We note the one study that exists investigating impacts of formalin on marine animal samples where the H-print index was not altered (Brown 2018, Polar Biology). Our trap samples were stored in the dark at room temperature, as cold storage was not required for the primary analyses of the trap material as analyzed by Lalande et al. (2020). An analysis by Cabedo-Sanz et al. (2016, Organic Geochemistry) showed that light, but not temperature led to degradation of HBIs. However, given the preservation of these samples (as opposed to storage of frozen/freeze dried/oven dried surface sediments or in solvent/dry HBI extracts), there are no existing studies that have explicitly addressed the impact that storage temperature has on preserved samples. We can confidently say that light degradation would not have been an issue here. 

Line 141 – Limoges et al 2018 indicate that H. spicula, not H. crucigeroides is an IP25 producer. In the Brown et al study, the authors did not distinguish between these two species. Add reference.

The text addresses “sympagic HBI producers” (IP25 and HBI II). While Limoges’ work showed H. crucigeroides was not producing IP25, we do know whether the species produces HBI II. Therefore, for clarity, we have changed the text to read “H. crucigeroides and H. spicula” with the addition of Limoges citation, rather than “and/or”.

Lines 142-144 – As mentioned earlier, the limitation is not the use of microscopy per se. It is the use of fresh samples/Utermohl method. If instead, cleaned frustules were examined at 1000X resolution with phase contrast it would have been possible to identify most of the species present.

Again, we thank the reviewer for the suggestion but refer to our previous comments that noted the limitations of the methodologies used by Lalande et al. 2020, and how we have provided this information more clearly in the text. 

Fig. 2 – sediment sampling locations “were selected” instead of “occurred”. 

Revised.

Lines 295 – Do you mean the opposite – i.e. concentrations decreased in late July? 

Actually, while the sea ice concentration decreases in July, it does not fall below 15% (the criteria for open water) and then it increases again to 60%. We rephrased the text here to ‘Some sea ice ( >15%) however remained present above the sediment trap until the end of deployment.

(See Reviewer #2 comment as well. The axis in Fig 3 was modified for clarity and a blue dashed line added for 15% sea ice concentration).

Lines 316-317 – Here it would be good to see fluxes normalized by TOC. Could the apparent decline in IP25 actually reflect an increase in total flux rates? 

After discussions among the coauthors, we concluded that the HBIs should not be normalized to TOC. Almost all prior studies using HBI fluxes have not normalized data in this manner (see Bai et al 2019, Lalande et al 2016, Fahl and Stein 2012 and recent review by Belt 2018). This is also a standard practice for surface sediments/down core studies, and followed that precedent here. Brown et al. 2016 (MEPS) presented HBI/POC values but this would be another context to make comparisons. 

Fig. 3 – It is important to add to the figure legend and the figure itself what year(s) the trap data cover (2015-2016). I don’t understand what “pieces” stands for – fragments? If so, why not include these in the spp. counts? And how large fragments were considered =1?

Revised figure legend and caption to include 2015 and 2016. 

“Pieces” were fragments. They were shown simply as an indicator of their presence. However, we have removed Gyrosigma/Pleurosigma/Haslea pieces and G. tenuirstrom fluxes since they were minor contributions, and do not change IP25 concentrations. 

We also removed Rhizosolenia fragments. However, these cells were a significant contribution to the flux and were often present as fragments when no intact cells with chloroplasts could be detected. As a result of removing all fragment fluxes, the correlation between HBI III and Rhizosolenia was weakened. We also tested the correlation of HBI III with two exclusively pelagic species (Chaetoceros and Thalassosira). There were no significant correlations found. We decided not to add this to the results. The fact remains that the HBI III attribution to a pelagic source remains inconclusive in this study.

The Pearson correlation table (Table 4) and associated text has been revised and the fragments have been removed from Fig 3.

Figs. 4 and 5 Legend – it should be clearly stated in the legend that sampling stations are not the same for each year. Sampling sites are not easy to see as they are plotted, and at a quick glance the figure could be mis-interpreted as showing a southward expansion of HBIs over time. 

Revised figure captions. 

Lines 668-670 – Is this supported by the data? 

We cite this study because it provides context for the current bloom dynamics in the N Bering Sea as dramatic changes in ice cover occur. We have re-written the text, particularly in the prior paragraph so it is clear that the citation is not related to the HBI III data, but rather to provide a possible mechanism for why the proportions are so different than in the Chukchi Sea. 

Lines 680-681 – explain what evidence you have to support this – not clear what data are behind the assumption that this is “a likely but minimal source”

We have added a statement on the rapid sinking of diatom aggregates that would likely prevent substantial HBI advection and contributions from drifting sea ice. 

Lines 688-690 – resting spores are dense and primed for sinking into the seafloor. It is more likely that the blooms are seeded from sediments than from the water column. Unless the life-cycle of M arctica is well studied, it is speculative to assume they survive in the water column.

Revised to reflect the reviewer’s suggestion. 

Lines 690-691 – this statement seems rather odd. Of course sea ice species have strategies to persist when sea ice is not present, as do all other aquatic protists. Dormancy is a wide-spread strategy in protists. Ellegaard and Ribeiro 2018 review the phenomenon of long-term dormancy of microalgae in aquatic “seed banks” – article published in Biological Reviews.

Reworded this sentence and clarified the presence of M. arctica resting spores exemplifies this strategy as noted in Ellegaard and Ribiero 2018 (citation added). We also added that perhaps the resuspension of Haslea and Pleurosigma resting cells contribute to the winter IP25 flux.

Lines 692-694 – very interesting finding – what species were kept in the laboratory? As T. Brown is also an author here, I suppose you could refer to it as “our own unpublished results” rather than a personal comm.

Added the species (H. crucigeroides and H. vitrea) and cited as unpublished data. T. Brown will not be publishing this work. 

Conclusions – the entire first paragraph of the conclusions should be moved/merged into the discussion section. 

Moved to beginning of discussion

I encourage the authors to consider what are the implications of their findings – back to the stated goals at the end of the introduction. For example, why and how may “lipid biomarkers serve as an integrating tool to better understand and monitor the rapid changes occurring in this ecosystem”? What are the potentials and limitations? And what issues need to be taken into account that are specific to this region, but perhaps do not apply to other parts of the Arctic?

We have added a few sentences that highlight the fact that the Pacific Arctic is one of the world’s most productive ocean ecosystems and is different from the rest of the Arctic in this respect. Limitations were noted with regards to the proxies not adequately capturing the communities present and the potential to supplement the existing measurements that attempt to quantify ice algae contributions without source specificity that HBIs offer. 

Reviewer #2: This paper presents highly branched isoprenoid (HBI) biomarker, including IP25, data and diatom data from a sediment trap in the Chukchi Sea and a suite of surface sediments across the Chukchi and Bering seas. It is among the first, perhaps the first, to report such data. This data is used to address the question of how much primary productivity occurs in ice covered waters. The authors propose a model for sea ice and pelagic diatom productivity and deposition across this highly productive region. It is an important paper that nicely summarizes the research and understanding of phytoplankton and sympagic algae.

I can’t comment on the biomarker/HBI methods, but I hope another reviewer was asked to look at this paper who is an HBI expert. I know that these methods have been tricky for some to properly emulate.

This is a well-written paper. The discussion is a bit long and could perhaps be shortened by editing and reorganizing the content that is currently on pages 22-25. But, I don’t have any significant comments or concerns. Some care needs to be given to the figures, which are quite pixelated in the pdf version of the manuscript. Several need additional annotations or the figure caption doesn’t match what is shown in the figure. 

All figures were checked for resolution and resized to hopefully resolve the pixilation issues. Other revisions to figure captions are addressed in comments below.

Minor line by line comments:

Line 100: Refer to Fig. 2

Revised.

Line 125: I was struck by how high the salinity was adjusted to in the collection cups. Is there a reason for this high salinity?

Revised to explain that the purpose of the high salinity water is to retain deposited material in the open sample cup until the trap rotates to a new cup. This has no impact on the diatoms or HBIs.

Lines 190-197: An additional sentence describing how you expect the 137-Cs profile to be similar to DBO 4.6 or why a core 50 nm away is expected to be an adequate substitution would be helpful.

This manuscript follows on a much more detailed manuscript that presented data for 40 radiocesium profiles collected throughout the Chukchi shelf (Cooper and Grebmeier, 2018). We depended upon the information in this paper, which discussed the predominance of bioturbation and sedimentation as factors affecting profiles throughout most of the shelf. We have added a statement explaining the similarities in biological activity and sediment characteristics (TOC and grain size) at DBO4.6 and UTX13-23, which are significantly correlated with radiocesium activity in this region. While less than ideal sea conditions did not permit collection of multiple cores at DBO4.6, based upon the referenced paper, we have no expectation that the core collected for biomarkers at DBO4.6 reflects a different bioturbated deposition history than the core at UTX13-23 or at many other locations on the shelf. 

Line 296: I’m not sure why you say, “sea ice concentration never dropped below 15% before the end of the sediment trap deployment.” From figure 3, it looks like sea ice drops below 15% in July, peaks just above 15% for a brief moment and then is below 15% when the trap is recovered.

We think both reviewers may have been looking at the snow depth axis instead of sea ice concentration. To make this clearer, we have rephrased the text as described above in the response to Reviewer 1 and added a blue dashed line for 15% sea ice concentration on Fig 3. 

Line 322: It would be nice to remind the reader here that high H-print values indicate high pelagic contributions.

Added, “representing a mixed to pelagic diatom contribution”. Since H-Print values were as low as 48%, which does indicate a mixed composition, but reaching 70%, which is more pelagic. 

Line 337: The H-print is really the proportion of pelagic to sympagic, not the other way around. Also, it indicates higher contributions of pelagic diatoms, not necessarily greater periods of ice free waters. It might be a proxy for ice, but it’s really just measuring diatom contributions.

We have revised the Fig 3 caption to reflect this point. 

Lines 379-380: I suggest removing the words “and sea ice cover” and “greater periods of ice-free surface waters” because H-print really indicates the algal contribution not actually sea ice.

We have revised the Fig 5 caption with same wording as above for Fig 3.

Line 438: What does “clayish” mean? Clay is a textural term meaning grains smaller than 4 um (or < 2 um if you’re a soil scientist). Do you mean clay-rich? Silt and clay? Fine grained?

Revised to “fine-grained”

Lines 468-479: I think the authors overstate the need to be cautious here. Although I agree, that there is a reason to be cautious with IP25 because the proxy is really based on species that arguably are very minor, the results that are presented actually strengthen the interpretation that IP25 is an appropriate proxy for sea ice. I would suggest replacing the clause, “suggests the need for further studies before a final interpretation can be made.” with the opposite, “strengthens our interpretation.”

Revised per reviewer suggestion 

Line 499: “A coeval of HBI II and HIB III” is strange wording. Perhaps you mean, “An association between HBI II and HBI III”?

Accepted reviewer’s suggestion

Line 572: Since we don’t know why diatoms produce IP25, we don’t really know whether in places where there is an increase in IP25 if it’s an increase in the number/mass of those diatoms, or just some kind of environmental event that causes the diatoms to produce IP25. It’s probably a good idea to keep this in mind.

Based upon this suggestion, we have revised the manuscript by adding two statements. Immediately after this sentence, we suggest environmental parameters could be the cause, specifically due to nutrient limitation increasing HBI production. The recent study by Brown et al. 2020 is also introduced in the previous section. We state here that the reasons for HBI synthesis are still unknown but that nutrient limitation has been shown to increase HBI production ten-fold. 

Line 589: Please remind the reader whether you’re referring to larger or smaller grain size.

We were referring to larger grain sizes. Revised in the text. 

Line 597: I think you’re missing the word, “continues” between, “year round,” and, “early ice.”

Revised 

Line 636: I think you mean the 137Cs profile from core UTX 13-23, not DBO4.6. 

The reviewer was correct. Revised. 

Line 639-657: Although DBO4.6 has more bioturbation than NNE14, it likely still gets older as you increase in depth. There is slightly less 137-Cs at depth, and I suspect that if you were to core deeper, you’d see the loss of 137-Cs and reach quite old sediments. In strongly bioturbated regions, the 137-Cs peak is smeared, but not necessarily obliterated. I think that the increased sympagic signature at depth is also likely due to decreasing sea ice over the past few decades.

Yes, this is possible but we were cautious to make this conclusion based on the high degree of bioturbation. Based upon the reviewer comment, we revised to add a tentative statement that it is possible this increase of sympagic HBIs at depth could also be associated with sea ice declines. 

Figures:

Figure 2: It would be helpful to label the boxes SLIP, CHIR, SECS, NECS, and BARC, and also repeat the boxes on Figure 4.

We have revised to the reviewer’s suggestion. The boxes without labels were added to both Figures 4 and 5 for clarity. This also helps to address the comment by reviewer #1 with regards to regions that were not sampled in certain years. 

Figures 4 and 5: It’s really difficult to see the stations on these figures. Maybe make them a hair larger and colored solid black instead of grey. Also, it would be helpful to reproduce the boxes around the different regions (maybe remind us in the figure caption their names north to south?) since you refer to the regions in the text.

Revised as described above in addition to making the station symbols larger and darker. We also changed the layout to better visualize and hopefully reduce any pixilation. 

Figure 6: You reversed A and B in the figure/caption.

Correct, thank you for catching. Revised.

Figure 7: The dots need to be slightly bigger on these plots. It’s impossible to distinguish colors, especially on panel B, but also the boxes in the legend on panel A are very small. For example, I can’t tell the difference between the color for NECS and SECS. Maybe you could just label the box plot?

Resized this figure to hopefully alleviate this concern. 

Figure 10: I love this figure, but it’s impossible to read the text/labels in the molecular diagrams. In your figure caption, you label IP25 as red and HBI III as blue, but it appears to be the opposite in the figure. I’m also not sure what the scratch marks are on the underside of the ice in the Chukchi Sea Nov-Dec. Please describe what the brown and green shading indicates also (sea ice vs. pelagic diatoms?).

We removed the ‘scratch marks’ from the sea ice, which were intended to represent diatoms within sea ice but is not critical. The molecular structures were clarified and the colors (red and blue) were changed to brown/yellow and green to match the respective sources (ice algae (yellow) and phytoplankton (green). 

Table 1: It would be helpful to include the distance from the CEO sediment trap for each core location in this table. I’m also a little confused with the table and figure captions embedded in the text. Is line 187 part of the table caption? If so, then that’s fine. If not, you already said this earlier in the text.

We have added the distances from CEO. The line spacing of the table caption has been changed to clarify this text from the rest of the manuscript. 

Other Author Revisions:

- Updated the Lalande et al. 2020 reference, as it is now in press. 

- Added two statements regarding HBI synthesis and nutrient limitation due to a newly accepted manuscript by Brown et al. (2020).

---

## [Editor Report · Decision Letter 1]

18 Mar 2020

Seasonal and latitudinal variations in sea ice algae deposition in the Northern Bering and Chukchi Seas determined by algal biomarkers

PONE-D-19-34317R1

Dear Dr. Koch,

We are pleased to inform you that your manuscript has been judged scientifically suitable for publication and will be formally accepted for publication once it complies with all outstanding technical requirements.

With kind regards,

Christof Pearce

Academic Editor

PLOS ONE
---

## [Editor Report · Acceptance letter]

27 Mar 2020

PONE-D-19-34317R1 

Seasonal and latitudinal variations in sea ice algae deposition in the Northern Bering and Chukchi Seas determined by algal biomarkers 

Dear Dr. Koch:

I am pleased to inform you that your manuscript has been deemed suitable for publication in PLOS ONE. Congratulations! Your manuscript is now with our production department. 

With kind regards,

on behalf of

Dr. Christof Pearce 

Academic Editor

PLOS ONE